# The encoding of touch by somatotopically aligned dorsal column subdivisions

Josef Turecek[1], Brendan P. Lehnert[1] & David D. Ginty[1✉]

The somatosensory system decodes a range of tactile stimuli to generate a coherent sense of touch. Discriminative touch of the body depends on signals conveyed from peripheral mechanoreceptors to the brain through the spinal cord dorsal column and its brainstem target, the dorsal column nuclei (DCN)[1,2]. Models of somatosensation emphasize that fast-conducting low-threshold mechanoreceptors (LTMRs) innervating the skin drive the DCN[3,4]. However, postsynaptic dorsal column (PSDC) neurons within the spinal cord dorsal horn also collect mechanoreceptor signals and form a second major input to the DCN[5–7]. The significance of PSDC neurons and their contributions to the coding of touch have remained unclear since their discovery. Here we show that direct LTMR input to the DCN conveys vibrotactile stimuli with high temporal precision. Conversely, PSDC neurons primarily encode touch onset and the intensity of sustained contact into the high-force range. LTMR and PSDC signals topographically realign in the DCN to preserve precise spatial detail. Different DCN neuron subtypes have specialized responses that are generated by distinct combinations of LTMR and PSDC inputs. Thus, LTMR and PSDC subdivisions of the dorsal column encode different tactile features and differentially converge in the DCN to generate specific ascending sensory processing streams.

Fast-conducting LTMRs (Aβ-LTMRs) detect light mechanical forces acting on the skin and mediate discriminative touch[8–11]. Aβ-LTMR signals are rapidly conveyed from the periphery, and their axons ascend the dorsal column of the spinal cord and directly contact the DCN of the brainstem. From the DCN, mechanosensory information is relayed to multiple targets in higher brain regions. Most sensory information is conveyed from the DCN to the somatosensory cortex through a prominent projection to the somatosensory ventral posterolateral thalamus (VPL) for the conscious perception of touch. A separate, lesser-known population of DCN neurons relay tactile information to the external cortex of the inferior colliculus (IC)[12,13]. In this brain region, the information is integrated and contextualized with auditory information. Other populations of DCN neurons project to the olivocerebellar system[14,15] to coordinate motor adaptation. DCN neurons can also project to secondary thalamic nuclei[16] involved in the affective state and to the spinal cord and periaqueductal grey[14,16]. Thus, the DCN is a conduit of incoming mechanosensory signals and broadly connect mechanoreceptors in the periphery to several major brain areas[17].

Somatosensory coding in DCN neurons is heterogeneous[18–21], but how tactile signals are organized within the DCN and distributed to downstream targets remains unknown. Using mice, we sought to determine how sensory representations of the hindlimb are encoded at this early stage of the somatosensory hierarchy. To achieve this, we selectively recorded neuron subtypes in the DCN (the gracile nucleus; Extended Data Fig. 1) in mice using antidromic activation and optogenetic tagging.

Different DCN neuron types encoded distinct aspects of mechanosensory stimuli suited to their projection targets. VPL projection neurons (VPL-PNs) are the most abundant cell type in the DCN, outnumbering IC projection neurons (IC-PNs) and local inhibitory interneurons (VGAT-INs) with an estimated proportion of VPL-PN:IC-PN:VGAT-IN of 2:1:1 (ref. [13]). VPL-PNs had small excitatory receptive fields with large regions of surround suppression (Fig. 1a–c,p-q). These VPL-PNs could entrain their firing to mechanical vibration; however, for the majority, this entrainment was restricted to a narrow range of frequencies below 150 Hz (Fig. 1c,d). DCN neurons projecting to the IC could be classified into several subgroups (Extended Data Fig. 2), but most commonly had large and exclusively excitatory receptive fields that included the entire hindlimb (Fig. 1f–h,p). Unlike VPL-PNs, most IC-PNs could entrain their firing to mechanical vibration across a broad range of frequencies (Fig. 1h–i). Vibration of the hindlimb evoked precisely timed action potentials that were entrained and phase-locked to vibrations that ranged from 10 Hz up to 500 Hz, the highest frequency tested. Thus, neurons projecting to the VPL are tuned to convey finely detailed spatial information. By contrast, neurons projecting to the IC poorly encode spatial detail and are better suited to encode a broad range of mechanical vibrations that may be correlated with auditory stimuli. VGAT-INs had a wide range of receptive field sizes, lacked inhibitory surrounds and, unlike PNs, typically lacked spontaneous firing (Fig. 1k–n,p–r). All three DCN cell types rapidly adapted to step indentations at low forces (Fig. 1e,j,o). In contrast to other cell types, VPL-PNs more reliably encoded the static phase of sustained high-force indentation, substantially above forces at which rapidly adapting and slowly adapting Aβ-LTMRs plateau[22–25] (Fig. 1e).

We next addressed how the specific response properties of VPL-PNs and IC-PNs are generated. Aβ-LTMR axons that travel through the dorsal

[1]Department of Neurobiology, Howard Hughes Medical Institute, Harvard Medical School, Boston, MA, USA. ✉e-mail: david_ginty@hms.harvard.edu

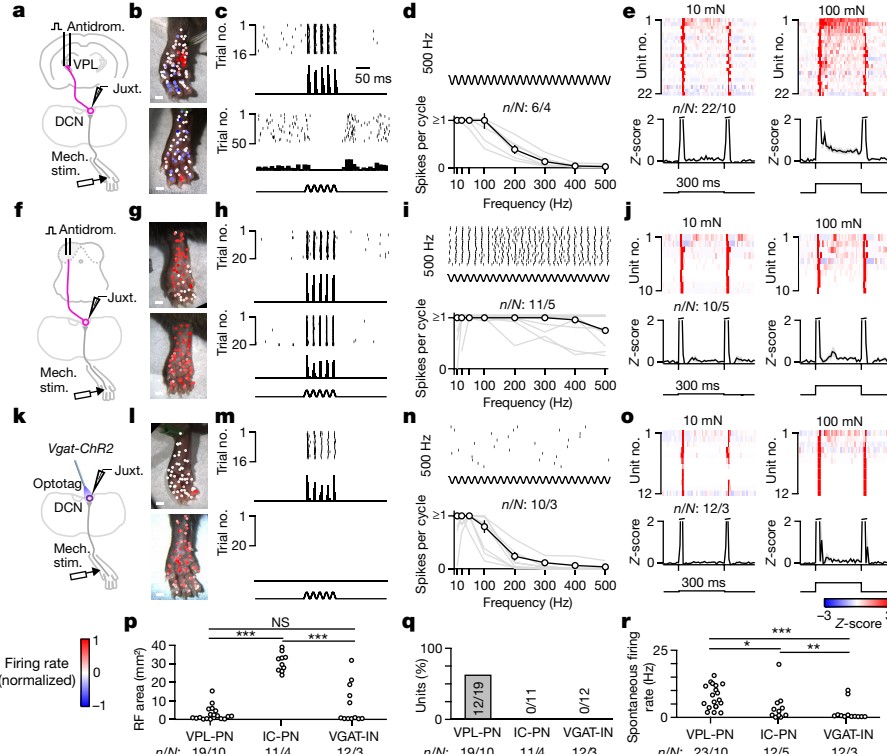

**Fig. 1 | Tactile features are directed to distinct targets in the somatosensory hierarchy. a**, Schematic of the experiment. Units in the DCN were recorded juxtacellularly (Juxt.) in urethane-anaesthetized mice. Stimulus electrodes were inserted into the VPL, and thalamic PNs (VPL-PNs) were identified through antidromic activation (Antidrom.) and collision testing. Response properties of units were then measured using a mechanical stimulator (Mech. stim.) with a 1 mm probe tip. **b**, Receptive fields of two different VPL-PNs. Each point is a single trial colour-coded by the normalized firing rate of the unit in response to 100 ms 50 Hz vibration (10–20 mN). **c**, Example trials from **b** with a raster and a histogram for two locations of the receptive field for the same unit: excitatory receptive field (top; 1 ms bins) and inhibitory surround (bottom; 10 ms bins). **d**, Top, Example raster of VPL-PN responses to 500 Hz vibration in the excitatory receptive field. Bottom, Vibration tuning of all VPL-PNs with the mean across

units ± s.e.m. (black) and individual units (grey). **e**, Average responses (*Z*-scored firing rate) to step indentation in VPL-PN units (top) and mean ± s.e.m. of VPL-PN units (bottom) for 10 mN (left) and 100 mN (right). Mice received 300 ms indentations delivered to the centre of excitatory receptive fields (10 ms bins). **f**–**j**, Same as **a**–**e**, but for units identified as projecting to the IC. **k**–**o**, Same as **a**–**e**, but for optotagged local inhibitory interneurons (VGAT-INs). **p**, Excitatory receptive field (RF) size area for all identified units. Kolmogorov–Smirnov (K-S) test: VPL-PN versus IC-PN, $P < 0.001$; IC-PN versus VGAT-IN, $P < 0.001$; VPL-PN versus VGAT-IN, $P = 0.4$. NS, not significant. **q**, Percentage of DCN cell types with detected inhibitory surrounds. **r**, Spontaneous firing rate of all units for each cell type. K–S test: VPL-PN versus IC-PN, $P = 0.012$; IC-PN versus VGAT-IN, $P = 0.008$; VPL-PN versus VGAT-IN, $P < 0.001$. Number of experiments are units/animals (*n*/*N*). Scale bars, 1 mm (**b**,**g**,**l**).

column form the 'direct' dorsal column pathway from the skin to the DCN and synapse directly on DCN PNs and interneurons. However, PSDC neurons of the spinal cord, which receive input from a broad array of somatosensory neuron subtypes, including Aβ-LTMRs, also project to the DCN through an 'indirect' dorsal column pathway that exists across mammals, including primates[2,5,7,26–31]. PSDC neurons constitute up to 40% of the axons ascending the dorsal column[6] (Extended Data Fig. 3), but the function of PSDC neurons and their contribution to somatosensory representations in the DCN and higher brain regions have remained unclear since their discovery. We proposed that direct and indirect dorsal column pathway projections specifically contribute to the distinct tuning features of DCN neuron subtypes.

To isolate the functions of direct and indirect dorsal column inputs to DCN neuron responses, we used the light-activated chloride channel ACR1 to reversibly silence axon terminals of ascending inputs. We first generated *Cdx2*<sup>cre</sup>*;Rosa26*<sup>LSL-Acr1</sup> mice to express *Acr1* in all neurons below the neck. This enabled reversible silencing of both primary sensory (direct pathway) and PSDC (indirect pathway) neurons that provide input to the DCN (Fig. 2a and Methods). We transiently silenced axon terminals in the DCN by preceding mechanical stimuli with brief, 300–400 ms light ramps (Extended Data Fig. 4), which was optimal for suppressing excitatory inputs (Methods). Mechanical stimuli were delivered in the final 100–200 ms of application of light. Using this strategy to silence both the direct and indirect dorsal

column pathway inputs abolished almost all DCN responses to vibration and low-force step indentation of the hindlimb (Fig. 2b–e). We next selectively silenced all direct dorsal column pathway (Aβ-LTMR) input using *Avil*<sup>cre</sup>*;Rosa26*<sup>LSL-Acr1</sup> mice. This mouse model enabled us to determine how the indirect pathway contributes to responses in individual DCN neurons (Fig. 2f). When light ramps were applied to silence Aβ-LTMR axon terminals in the DCN to block direct pathway inputs, the amplitude of responses was reduced but not eliminated. The indirect pathway was especially able to convey signals from low-frequency (10 Hz) mechanical stimuli to generate responses in the DCN (Fig. 2g,h). Silencing LTMR inputs also affected the response to step indentations, but the indirect pathway was able to reliably convey the onset and, in most units, the offset of low-threshold step indentation stimuli (Fig. 2i–j). Conversely, indirect pathway input failed to evoke firing in response to 50 and 300 Hz vibratory stimuli. These findings were consistent across randomly sampled DCN neurons and in identified VPL-PNs and IC-PNs. This result suggested that the direct pathway drives high-frequency vibration across DCN cell types (Extended Data Fig. 4). Thus, high-frequency time-varying light-touch stimuli such as vibration are exclusively encoded by the direct dorsal column pathway. By contrast, both the direct and indirect dorsal column pathways contribute to responses to gentle or light skin displacement across DCN neurons. The indirect pathway can also contribute to low-frequency (10 Hz) vibration responses.

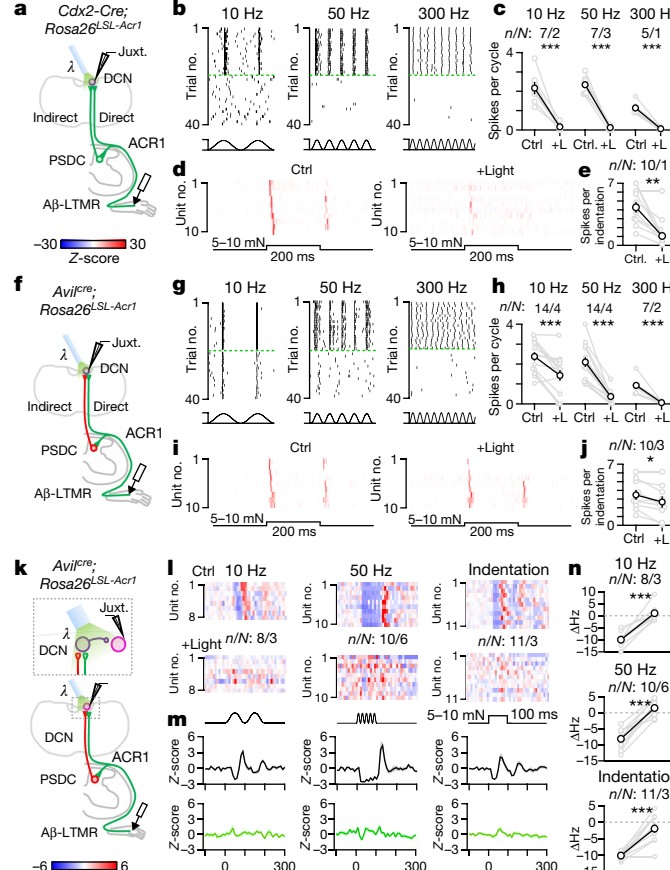

**Fig. 2 | The direct dorsal column pathway conveys high-frequency vibration and fine spatial information to the DCN. a**, Schematic for silencing all ascending input to the DCN (direct and indirect). ACR1 was expressed in all spinal cord and sensory neurons below C2 in *Cdx2-Cre;Rosa26^LSL-Acr1^* mice. Light was applied to terminals in the DCN to silence inputs in interleaved trials. Then 300–400 ms ramps of light (0.46 mW mm^−2^) preceded 100–200 ms of mechanical stimuli. **b**, Rasters of single random DCN unit responses to different vibration frequencies (10–20 mN) at baseline or when silencing all input. Silencing trials are sorted and separated by green broken lines. **c**, Average spikes per cycle during vibration at baseline (Ctrl) or when silencing (+Light (+L)) all input for individual units (grey) and mean ± s.e.m. (black). Paired *t*-test: 10 Hz, *P* < 0.001; 50 Hz, *P* < 0.001; 300 Hz, *P* = 0.001. **d**, Average histograms (*Z*-scored firing rate) for indentation (5–10 mN, 200 ms) at baseline (left) and when silencing all input (right). **e**, Average spikes per indentation at baseline and when silencing ascending input. Paired *t*-test: *P* = 0.002. **f**, Schematic for silencing the direct pathway. ACR1 was expressed in sensory neurons in *Avil^cre^;Rosa26^LSL-Acr1^* mice. The remaining DCN responses are mediated by PSDC neurons. **g–j**, Same as **b–e**, but silencing the direct pathway only. Paired *t*-test: 10 Hz, *P* < 0.001; 50 Hz, *P* < 0.001; 300 Hz, *P* = 0.001; indentation, *P* = 0.011. **k–n**, Schematic for silencing the direct pathway during mechanical activation of inhibitory surrounds in random units. ACR1 was expressed in sensory neurons in *Avil^cre^;Rosa26^LSL-Acr1^* mice. **l**, Average response of DCN units to stimulation of their inhibitory surrounds: vibration and indentation at baseline (top) or silencing the direct pathway (bottom). **m**, Mean ± s.e.m. of inhibitory responses across DCN units for vibration and indentation at baseline (top) or silencing the direct pathway (bottom). **n**, Change in firing rate for units when stimulating the inhibitory surround at baseline or silencing the direct pathway. Individual units (grey) and mean ± s.e.m. (black). Paired *t*-test: 10 Hz, *P* < 0.001; 50 Hz, *P* < 0.001; indentation, *P* < 0.001. Number of experiments are units/animals (*n/N*). All *t*-tests are two-sided.

VPL-PNs can encode spatial information partly because of their prominent inhibitory surround receptive fields. Therefore we examined how the direct and indirect dorsal column pathways contribute to surround inhibition of these PNs (Fig. 2k). Spontaneously active

VPL-PNs were effectively inhibited by applying brief vibratory stimuli or indentation to areas outside their excitatory receptive field. (Fig. 2l,m). Silencing Aβ-LTMR axon terminals in the DCN almost completely abolished inhibitory surrounds generated by 10 or 50 Hz vibratory stimuli and indentation (Fig. 2l–n). Thus, the direct dorsal column pathway is the primary driver of inhibitory surround receptive fields in VPL-PNs generated by these stimuli.

We next asked how PSDC neurons and the indirect pathway contribute to DCN representations of other features of mechanical stimuli. There are currently no genetic tools that can selectively silence PSDC neurons. Thus, we used a pharmacological approach to block the indirect dorsal column pathway (Fig. 3a). Application of glutamate receptor antagonists directly to the dorsal surface of the lumbar cord effectively blocked excitatory synaptic transmission in the lumbar spinal cord dorsal horn (Extended Data Fig. 5). This in turn eliminated hindlimb-level PSDC neuron activation and contributions to responses recorded in the DCN. Glutamate receptor antagonists applied to the lower thoracic cord did not affect DCN responses to mechanical stimulation of the hindlimb (Extended Data Fig. 5). This result shows that glutamatergic transmission blockade was spatially restricted to the spinal cord region to which it was applied. Next we examined the effects of inhibiting fast excitatory transmission in the lumbar spinal cord. DCN neurons could still entrain and phase-lock their spiking to a broad range of vibration frequencies of mechanical stimuli applied to the hindlimb (Fig. 3b,c). This finding, which is consistent with results of the *Acr1* silencing experiments (Fig. 2h), indicates that Aβ-LTMR input through the direct dorsal column pathway underlies high-frequency vibration tuning in the DCN.

Another salient feature encoded by DCN neurons is stimulus intensity. We applied step indentations to the skin using blunt and smoothed probes (1 mm in diameter) that generated graded, compressive stimuli from low to high force ranges (1–300 mN). Although high forces were used, stimuli were applied over a wide area of skin. These stimuli were not noxious as they failed to evoke paw withdraw or pain-related behaviour in awake unrestrained animals, and often generated no observable reaction (Extended Data Fig. 6 and Supplementary Video 1). Under normal conditions, DCN neurons were highly sensitive to the onset and offset of low-force indentations. However, they also fired in response to sustained indentation of the skin, especially at high forces (Fig. 3d–f). High-force responses were prominent in VPL-PNs and, to a lesser extent, in IC-PNs (Fig. 1e,j). Sustained firing during maintained step indentations was proportional to the force applied. DCN neurons therefore not only detected the onset and offset of gentle stimuli but also encoded sustained mechanical stimuli across a broad range of intensities, into the high force range (Fig. 3f).

We next sought to determine the contribution of PSDC neurons to innocuous high-intensity stimuli. We performed recordings of DCN units that had receptive fields and spontaneous firing characteristic of VPL-PNs, the most abundant cell type in the DCN. When excitatory transmission was blocked in the spinal cord to suppress PSDC input, firing during the sustained phase of indentation was almost eliminated across all forces for most units (Fig. 3g–k). Thus, in the absence of PSDC input, DCN neurons could no longer encode the intensity of maintained stimuli (Fig. 3m). Moreover, attenuation of the PSDC input to the DCN increased the threshold of DCN neurons to the onset of step indentation (Fig. 3l). These findings suggest that PSDC neurons provide graded force information to the DCN, which enable responses to sustained high-intensity stimuli. At the same time, PSDC neurons contribute to the detection of gentle touch stimuli. Thus, PSDC neurons and the indirect dorsal column pathway are required for the wide dynamic range of intensity tuning in DCN neurons. This system enables the detection and encoding of a broad range of stimulus intensities.

The graded coding of intense stimuli in the DCN was also relayed upstream to middle VPL (mVPL) neurons. Both low-threshold sensitivity and sustained responses to high-threshold stimuli in the mVPL strongly depended on the DCN (Extended Data Fig. 7). Lesioning the DCN also altered spontaneous firing in the mVPL (Extended Data Fig. 7), although

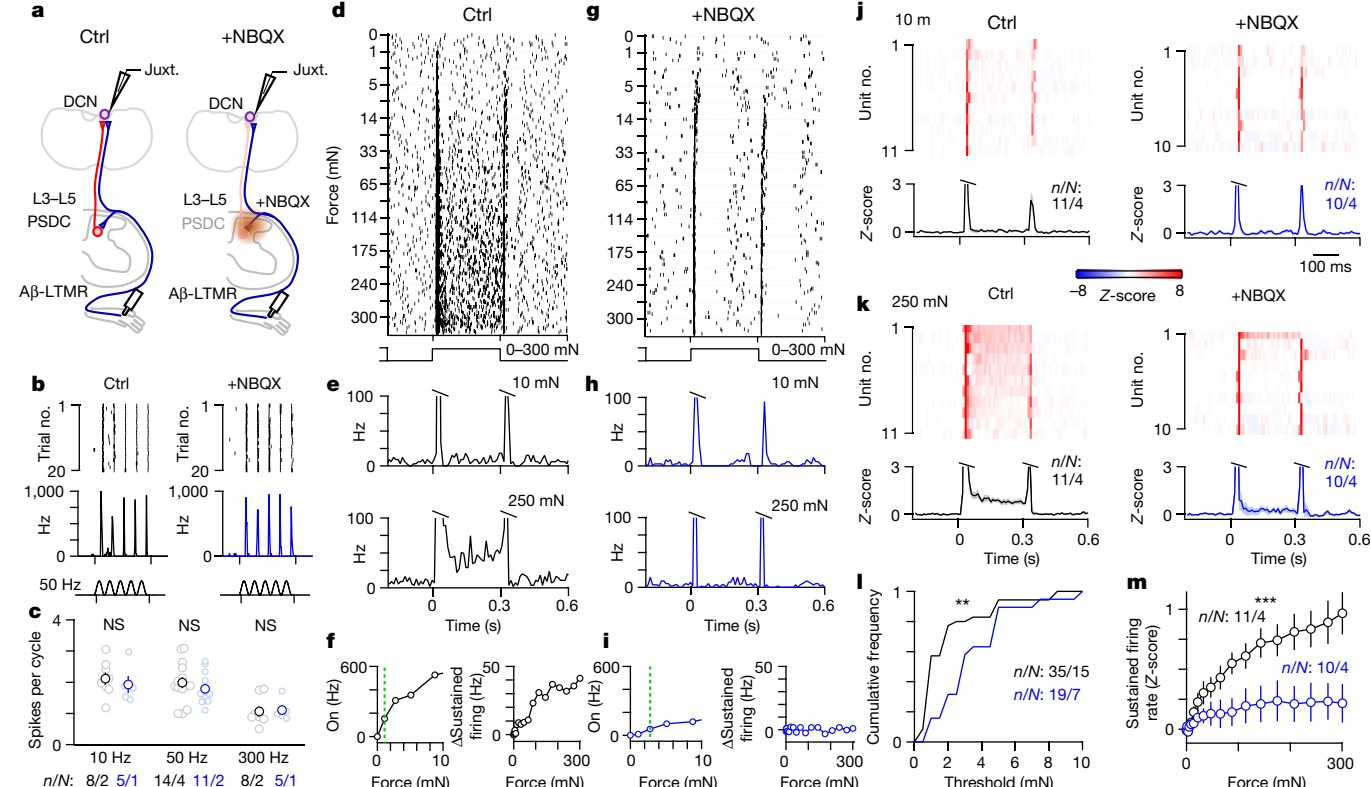

**Fig. 3 | PSDC neurons and the indirect dorsal column pathway mediate wide dynamic range force intensity tuning in the DCN. a**, Schematic for juxtacellular recordings from random units (vibration) or units with VPL-PN-like receptive fields (indentation) in the DCN. A L3–L5 laminectomy and durotomy was performed to apply MK-801 (10 mM, 10 μl, 5 min) followed by NBQX (10 mM, 20 μl). **b**, Single-unit responses to 50 Hz vibration (10–15 mN) with raster (top) and histogram (bottom) under control conditions (left) and different units with inhibitors applied to the spinal cord (right). Shown in 1 ms bins. **c**, Summary of vibration responses for all units recorded under control (black) or with inhibitors (blue), individual units (light) and mean ± s.e.m. (dark). K–S test: 10 Hz, *P* = 0.51; 50 Hz, *P* = 0.16; 300 Hz, *P* = 0.51. **d**, Raster of responses of single units to 300 ms of indentation at different forces (0–300 mN; 0–395 kPa) using blunt probes (1 mm in diameter). Trials were originally interleaved but are sorted here by force for presentation. High forces are innocuous in awake

animals (Extended Data Fig. 6). **e**, Histogram for a single unit in **d**–**f** for 10 (top) and 250 mN (bottom), 10-ms bins. **f**, Left, Maximum on-response (0–20 ms) compared to indentation force for the unit in **d**. The green line indicates threshold. Right, Average sustained firing (100–300 ms) compared to force for the unit in **d**. **g**–**i**, Same as **d**–**f**, but for a different unit treated with inhibitors. **j**, Average histograms of responses for all units to 10 mN of step indentation (top) and mean ± s.e.m. across units (bottom) under control conditions (left). Responses to different DCN units with inhibitors in the spinal cord (right). Shown in 10 ms bins. **k**, Same as **j**, but for 250 mN of indentation. **l**, Cumulative histogram of on-response threshold for all units in control (black) and for inhibited units (blue). K–S test: *P* = 0.002. **m**, Mean ± s.e.m. sustained firing rate across units under control conditions (black) and when inhibited (blue). K–S test: *P* < 0.001. Number of experiments are units/animals (*n/N*).

this manipulation will also affect the corticospinal tract. Many neurons in the DCN are spontaneously active, but lesioning the dorsal column did not have major effects on spontaneous firing in randomly recorded DCN neurons (*P* = 0.93, Kolmogorov–Smirnov test; 16 units, 2 mice). This result suggested that PSDC neurons do not drive spontaneous firing in the DCN.

The observation that DCN neurons encode high-intensity indentation stimuli is noteworthy because most Aβ-LTMRs are thought to saturate their firing at relatively low indentation forces[22–25]. As most of the high-force responses of DCN neurons during sustained indentations are mediated by PSDC neurons, we considered the possibility that PSDC neurons transmit signals emanating from both LTMRs and high-threshold mechanoreceptors (HTMRs), which typically do not project directly to the DCN through the direct dorsal column pathway[32]. To activate HTMRs without concurrent activation of LTMRs, we expressed the light-activated cation channel ReaChR in somatosensory neurons expressing *Calca* (CGRP) using *Calca-FlpE; Rosa26^{FSF-ReaChR}* mice and applied light directly to the skin of the hindlimb (Fig. 4a). Among the sensory neurons that express *Calca* are A-fibre HTMRs, C-fibre HTMRs, thermoreceptors and polymodal C-fibre neurons[33–36]. As described above, VPL-PNs responded rapidly to strong but innocuous mechanical step indentations of the skin with additional long latency spikes

(Fig. 4b,d and Extended Data Fig. 8). Optical excitation of *Calca*⁺ HTMRs within the same area of skin triggered firing in VPL-PNs with fast latencies that were consistent with A-fibre activation, but were longer than those evoked by mechanical stimuli (Fig. 4b–e). These rapid optogenetically evoked responses were sometimes followed by a second and much longer latency burst (Fig. 4e). These two temporal components of firing were consistent with the activation of intermediate-conducting and slow-conducting Aδ-fibre and C-fibre neurons known to express *Calca*[34]. Notably, almost all VPL-PNs fired strongly and consistently to optical activation of *Calca*⁺ endings in the skin (Fig. 4c,e). The slow latency of these responses was probably not due to slow opsin kinetics, as activation of all endings in the skin, including Aβ-LTMRs, generated fast latency responses (Extended Data Fig. 8). Previous work has shown that *Calca*⁺ neuron input to the DCN is sparse[37,38]. We observed few fibres in the DCN labelled by *Calca-FlpE*, and direct optical stimulation over the DCN failed to evoke firing in the same units that could be activated by optical stimulation of endings in the skin (Extended Data Fig. 8). Moreover, responses in the DCN evoked by activation of *Calca*⁺ endings in the skin depended on synaptic transmission in the spinal cord, as they were strongly attenuated by blockers of excitatory synaptic transmission applied to the spinal cord (Extended Data Fig. 8). Responses in the

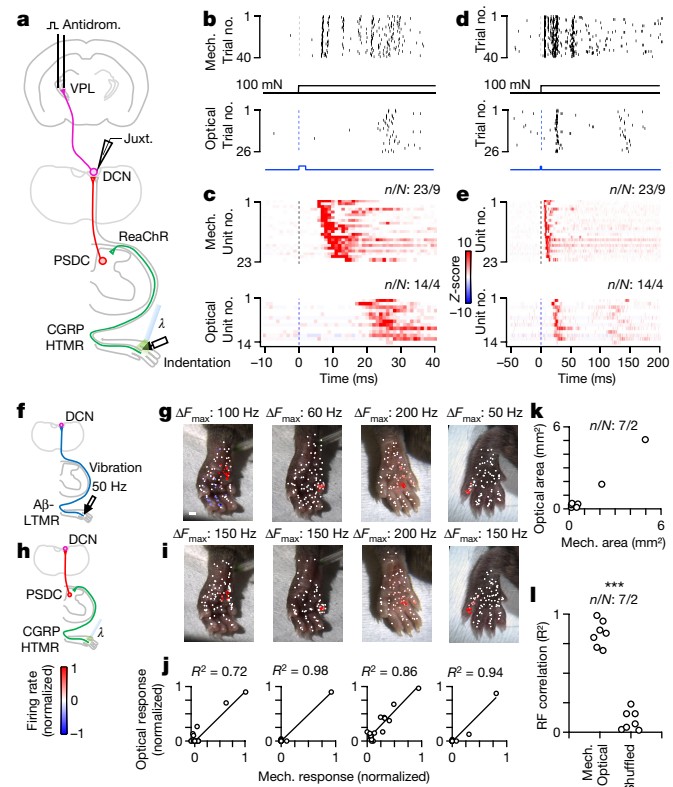

**Fig. 4 | Somatotopic convergence of the direct and indirect dorsal column pathways onto individual VPL-PNs. a**, Schematic of the experiment. VPL-PNs were identified with antidromic activation in *Calca-FlpE;Rosa26*^FSF-ReaChR animals. Pulses of light activated ReaChR in *Calca*+ HTMRs in the hindlimb. **b**, Raster of single VPL-PN unit responses to indentation (100 mN, 1 mm probe tip; top) and responses of the same unit to optical activation (2 ms, 60 mW mm⁻²) of HTMRs in skin (bottom). **c**, Histograms for all identified VPL-PNs for mechanical indentation (100 mN, top) and VPL-PN responses to optical activation of HTMRs in the skin (bottom). Shown in 1 ms bins. *Z*-score colour scale on the right. **d**–**e**, Same as **b**–**c**, but for a longer timescale. **f**, Schematic of mechanical (direct) stimulation. A 50 Hz 100 ms vibration (10–20 mN) was applied to different points on the skin for putative VPL-PNs in *Calca-FlpE;Rosa26*^FSF-ReaChR animals. This stimulus activates the direct pathway. **g**, Vibration receptive field maps for four VPL-PNs. Each point is a single trial colour-coded by its normalized mechanically evoked firing rate. Colour scale on the bottom left. Scale bar, 1 mm. **h**, Schematic of optical (indirect) stimulation. For the same units in **g**, light pulses (5 pulses at 0.5 Hz, 2 ms, 60 mW mm⁻²) were applied to different points on the skin. This stimulus activates the indirect pathway. **i**, Optical receptive field for same units in **g**. Each point is a single trial colour-coded by its normalized optically evoked firing rate. **j**, Correlation of optical and mechanical responses. Each point is the average mechanically and optically evoked responses for a single region of skin. **k**, Area of optically evoked receptive field compared with the area of mechanically evoked receptive field for single units. Each point is one DCN unit. **l**, Correlation coefficients (*R*²) of optically and mechanically evoked receptive fields within individual units or *R*² of receptive fields shuffled between units. K–S test: *P* < 0.001. Number of experiments shown as units/ animals (*n*/*N*).

DCN were mediated by the dorsal column, as severing the dorsal column eliminated light-evoked responses in the DCN (Extended Data Fig. 8). These findings suggest that high-threshold information is relayed to the DCN by PSDC neurons that receive input from HTMRs, either directly or through local interneuron circuits within the spinal cord dorsal horn.

We also observed responses in IC-PNs evoked by stimulating *Calca*+ endings in the skin. However, they were weaker than those seen in VPL-PNs and were restricted to a subset of the receptive field that is more sensitive to low-frequency mechanical stimuli (Extended Data

Fig. 9). Activation of *Calca*+ endings failed to evoke inhibitory surrounds (Extended Data Fig. 9), which is consistent with the absence of sustained high-force responses in VGAT-INs (Fig. 1o).

Next we asked whether receptive fields of VPL-PNs are shaped by direct and indirect dorsal column pathway inputs. We measured the contributions of the direct and indirect dorsal column pathways to receptive fields of individual VPL-PNs. To do this, we determined Aβ-LTMR input contributions to VPL-PN receptive fields by using vibratory stimuli, as high-frequency vibration is encoded by the direct pathway (Fig. 4f,g). In the same experiment, we determined PSDC input contributions to receptive fields of the same VPL-PN neurons using optical activation of *Calca*+ endings in the skin. This is because *Calca*+ neuron inputs to the DCN are conveyed solely by the indirect pathway (Fig. 4h,i). The receptive fields of the direct and indirect dorsal column pathway inputs onto VPL-PNs were highly aligned (Fig. 4g,i,j–l). Responses to gentle vibration were typically restricted to a single digit or to one to two pads. Moreover, optical activation of HTMRs evoked firing only in areas that were sensitive to vibratory stimuli, restricted to the same single digit or to one to two pads. These findings suggest that there is an elaborate somatotopic alignment of the periphery, spinal cord and DCN. That is, Aβ-LTMRs and HTMRs that innervate a small area of skin project into the central nervous system and diverge, with Aβ-LTMRs projecting through the dorsal column directly to the DCN, and both LTMRs and HTMRs activating PSDC neurons in the spinal cord dorsal horn. Aβ-LTMRs (direct pathway) and PSDC neurons (indirect pathway) with similar receptive fields then re-converge within the DCN to enable representation of a broad array of tactile features for a single, small area of skin.

## Discussion

The dorsal column system enables the perception of a rich array of tactile features[2–4,11]. Models of discriminative touch have primarily focused on the roles of LTMR subtypes and their direct dorsal column pathway projections in the creation of these representations[4,9,11]. Here we demonstrated that PSDC neurons are a crucial component of a brainstem circuit that transforms ascending tactile inputs to produce specialized tactile representations.

Our findings provide support for a new model of the dorsal column discriminative touch pathway (Extended Data Fig. 10). Aβ-LTMRs and the direct dorsal column pathway underlie vibration tuning, whereas PSDC neurons and the indirect dorsal column convey the intensity of sustained stimuli. Both pathways detect the onset of stimuli and low-threshold responses, together providing high sensitivity. Notably, these two components of the dorsal column pathway differentially converge on distinct DCN-PN subtypes to generate specific combinations of response properties in different sensory streams. Information conveyed to the primary somatosensory cortex through VPL-PNs emphasize spatial detail, moderate-to-low frequency vibration (<150 Hz) and sustained stimulus intensity. These features are probably generated by prominent input from both Aβ-LTMRs and PSDC neurons. Tactile signals conveyed to the IC by IC-PNs encode broadband vibratory information (10–500 Hz) rather than spatial detail, and are probably mainly driven by direct LTMRs, especially Meissner corpuscles (innervated by type 1 rapidly adapting Aβ-LTMRs) and Pacinian corpuscles (innervated by type 2 rapidly adapting Aβ-LTMRs). We found that LTMR input generates much of the surround inhibition in VPL-PNs and that VGAT-INs do not fire in response to sustained high-force stimuli, which suggests that they do not receive HTMR input through PSDCs. We also found that PSDC neurons do not have a major role in surround inhibition. However, it remains possible that they are involved in other aspects of mechanically evoked or tonic inhibition that we did not measure. Many of the DCN response properties reported here can be observed in both anaesthetized and non-anaesthetized conditions[39–41], but PSDC neurons may have additional roles in awake behaving animals[42,43]. Similar to the division of LTMRs into subtypes, there may also be

physiologically distinct subtypes of PSDC neurons given their various contributions to the DCN described here and their heterogeneous response properties in cats[5]. Future work will address potential PSDC subdivisions, how features are represented across a broader range of DCN PN subtypes, such as those that project to the cerebellum and higher-order thalamic nuclei, and how PSDC neurons contribute to touch in different behaviours.

We observed distinct somatotopic alignment of the direct and indirect dorsal column pathway inputs to VPL-PNs in the DCN. This alignment enables rich representations in VPL-PNs without compromising spatial detail. The development of this somatotopy probably requires complex coordination between primary sensory neurons, including both LTMRs and HTMRs, spinal cord dorsal horn circuitry and the DCN. Developmental activity, either spontaneous or mechanically evoked, may play an essential part in organizing this system, as it does in other sensory systems[44,45]. Moreover, adult PSDC neurons exhibit strong receptive field plasticity[46,47], which raises the possibility that signal propagation through the indirect dorsal column pathway can be modified by sensory experience. Thus, subdivisions of the dorsal column pathway may not only expand the capacity for coding tactile features across DCN output pathways but also introduce hard-wired and flexible components to the repertoire of mechanosensory representation in the earliest stages of the sensory hierarchy.

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

## Methods

### Animals

All experimental procedures were approved by the Harvard Medical School Institutional Care and Use Committee and were performed in compliance with the Guide for Animal Care and Use of Laboratory Animals. Animals were housed in a temperature-controlled and humidity-controlled facility and were maintained on a 12–12 h dark–light cycle. All experiments were performed on adult animals (aged >5 weeks) of both sexes. The following mouse lines were used: C57Bl/J6; $Cdx2^{cre}$ (ref. [48]); $Calca\text{-}FlpE$ (ref. [49]); $Avil^{FlpO}$ (ref. [49]); $Avil^{cre}$ (ref. [50]); $Rosa26^{LSL\text{-}Acr1}$ (ref. [51]); and $Rosa26^{FSF\text{-}ReaChR}$ (derived from ref. [52]). Animals were maintained on mixed C57Bl/J6 and 129S1/SvImJ backgrounds. C57Bl/J6 mice were obtained from The Jackson Laboratory.

Experiments were not blinded because mice and treatments were easily identifiable as experiments were performed. Whenever applicable, animals were randomized to different treatment conditions. Sample sizes were not predetermined.

### Juxtacellular recordings

All recordings were performed in urethane-anaesthetized mice. Adult (aged >5 weeks) animals were anaesthetized with urethane (1.5 g kg$^{-1}$) and placed on a heating pad. The head and neck were shaved and local anaesthesia (lidocaine HCl, 2%) was administered to the scalp and neck. An incision was made in the skin, and the muscle of the dorsal aspect of the neck was cut and moved aside to expose the brainstem. A head plate was attached to the skull using dental cement and the head was fixed to a custom-built frame. The dura overlying the brainstem was cut and small fragments of the occipital bone were removed. In some cases in which optical access to the DCN was required, small amounts of the posterior vermis of the cerebellum were aspirated to expose the DCN. A glass electrode filled with saline (2–3 MΩ) was put in place 200–300 μm above the gracile nucleus. The area was then flooded with low-melting point agarose dissolved in saline. After the agarose hardened, the electrode was advanced into the gracile. The electrode was guided to individual units and positioned to maximize the signal of a single unit. Recording quality and unit discrimination were continuously assessed using an audio monitor and online analysis of amplitude and spike waveforms. All recordings in the DCN were made in the hindlimb representation region of the gracile nucleus, and only units with receptive fields in the hindlimb were recorded. Recordings were targeted to the rostral–caudal level approximately where the gracile diverges bilaterally (see example in Extended Data Fig. 1) where units receiving input from the hindlimb digits and pads were most abundant. Although it has been reported that the gracile in rats, cats and primates is subdivided into 'core' and 'shell' regions, we were unable to detect clear organization in the mouse through electrophysiology experiments. Signals from single units were amplified using the ×100 AC differential amplification mode of a Multiclamp 700B instrument and sampled at 20 kHz using a Digidata 1550B controlled by Clampex 11 software (Molecular Devices). Signals were collected with an additional 20× gain, a 0.3 kHz high-pass filter and a 3 kHz Bessel filter.

In some experiments, DCN neurons were identified through antidromic activation of their axon in a target region. A craniotomy above the region of interest was performed. The head was levelled and bipolar electrodes (platinum–iridium 250 μm spacing, FHC) were lowered into the contralateral VPL of the thalamus (coordinates: 2 mm posterior to bregma, 2 mm lateral, 3.5 mm deep) or the contralateral IC (coordinates: electrode angled 45°, 4 mm posterior to bregma at tissue entry, 1 mm lateral, 1.7 mm deep). A single stimulus (60–200 μA) was applied every 3 s while searching for units in the DCN. A collision test was performed for all units that could be antidromically activated. Stimuli were triggered from spontaneous spikes, and experiments were only continued for units that passed collision testing. We found that units with reliable and precisely timed antidromic spikes (jitter < 1 ms) almost always passed a collision test. Units that failed collision testing had variable timing of activation and much longer latencies. Polysynaptic activation in the DCN was rarely detected by activating the IC, and often observed when stimulating the VPL. In a subset of experiments, once data collection was complete, the stimulus electrode was retracted from the stimulation site and coated with DiI (ThermoFisher). The electrode was then advanced back into the tissue and left in place for at least 5 min. To verify that the electrode was in the same position as before, units were antidromically activated again. Electrodes were then retracted and the animal was perfused as described below for anatomical verification of the stimulus electrode position. In another subset of experiments, following identification of a VPL-PN, the stimulus electrode was then retracted from the VPL and moved to coordinates of the posterior nucleus of the thalamus (coordinates: 2 mm posterior to bregma, 1.1 mm lateral, 3 mm deep). Units that were originally antidromically activated from the VPL failed to be activated by stimulation in the new location (number of experiments are units/animals: 3/3). Stimulation (60–200 μA) of the posterior nucleus also failed to evoke polysynaptic and multi-unit background activity in the DCN, as was often seen with VPL stimulation.

To record from inhibitory neurons in the DCN, we performed optical activation in *Vgat-ChR2* animals. Units were searched while applying 200–300 ms ramps of blue light (2–4 mW mm$^{-2}$) from an optic fibre (400 μm diameter, 0.39 NA) placed above the DCN. Light was delivered from a 470 nm LED (M470F3, Thorlabs). Ramps were used because pulses of light were found to generate short latency activation in most units. Most units in the DCN are glutamatergic, and we found that pulsed light drives strong synchronized GABAergic input to primary afferent terminals and stimulates them through the depolarizing action of GABA, thereby driving vesicle release onto excitatory projection neurons. When ramps were used, many units were instead silenced, as expected. Units considered optotagged were those that could be activated during ramps of light.

### Optical silencing

For optical silencing of ascending inputs in vivo, experiments were performed as described above, except dissection was performed in the dark under red-light illumination, as bright-white dissection light was capable of activating ACR1. All experiments were performed in animals homozygous for the $Rosa26^{LSL\text{-}Acr1}$ allele, as light was unable to fully silence inputs in heterozygous animals. Once preparation was complete, the electrode was positioned in place above the DCN along with a 400 μm diameter 0.39 NA fibre 1 mm above the DCN. The area was flooded with low-melting point agarose to embed the fibre and electrode. The electrode was then advanced into the DCN to record from single units.

Silencing trials were pseudorandomly interleaved with control trials. For silencing trials, a 300–400 ms ramp of light (0.2–0.8 mW mm$^{-2}$, 552 nm) was delivered to the surface of the DCN with mechanical stimuli delivered during the last 100–200 ms of the ramp. Light was delivered using a 552 nm LED (MINTF4, Thorlabs). We found that silencing became progressively ineffective after 500 ms of delivery of light, which was possibly due to effects of prolonged depolarization of the terminals.

Optical silencing could only be performed successfully for gentle stimuli (≤ 20–30 mN). In *Cdx2-Cre;Rosa26$^{LSL\text{-}Acr1}$* animals, light was unable to fully silence inputs to the DCN when delivering intense indentations. Silencing using ACR1 may substantially suppress vesicle release from ascending inputs. However, the simultaneous and ongoing activation of many inputs may still allow postsynaptic DCN neurons to reach threshold despite a large reduction in the amount of synaptic drive. Thus, experiments were limited to gentle indentation and vibratory stimuli.

### Mechanical and optical stimuli

We selected units that were primarily responsive to stimulation of the hindlimb. All mechanical stimuli were generated by a DC motor

with an arm connected to a blunt and smoothed acrylic probe tip that was 1 mm in diameter. The use of a large diameter probe tip with smoothed edges allowed the delivery of high forces that that did not evoke paw withdraw in awake animals (Extended Data Fig. 6). We did not observe visible damage to the skin following high-force stimuli in awake or anaesthetized animals. The motor was driven by a custom-built current supply controlled by a data acquisition board (Digidata 1550B, Molecular Devices). Step indentation forces were calibrated using a fine scale. For experiments measuring responses to various forces, the probe tip was positioned in a resting state on the skin surface, and indentations of incrementally increasing forces were applied every 3–10 s. Once the maximal force was reached, the force was reset to zero and was again incrementally increased. For experiments measuring responses to vibration, vibratory stimuli were applied at similar forces (10–20 mN) across frequencies. DCN units that did not respond to Pacinian-range vibration frequencies also did not respond when vibrations were delivered at higher forces.

The stimulator was attached to an articulating arm that could be moved by hand and would remain in position. For receptive field mapping, the probe tip was manually positioned over a single location of the limb where it remained in place. A trial was then initiated to deliver a brief vibration (50 Hz, 100 ms, 10–20 mN). The experiment was performed under a stereoscope equipped with a CCD (BlackFly S BFS-U3-04S2C, Flir) operated by Spinview 2.3.0.77, and was triggered to capture images for each trial. This process was repeated until enough trials were acquired (60–100) to generate a receptive field map of the entire hindlimb. For experiments measuring indentation responses at various forces, the stimulator was firmly secured to a 0.5-inch heavy post to prevent relocation or repositioning when applying high forces.

Optical stimulation was performed on the skin or DCN using a 200 μm or 400 μm diameter 0.39 NA fibre coupled to a 554 nm LED (MINTF4, Thorlabs). For skin stimulation, the fibre tip was held in place with an articulating arm and manually moved into position for each trial, as performed for mechanical stimulation. For optical receptive field mapping, five pulses (2–5 ms duration, 60 mW mm$^{-2}$) were delivered to the skin at 1 Hz for each trial.

## Multielectrode array recordings

Recordings in the thalamus were made in the mVPL (2 mm bregma, 2 mm lateral, 3.5 mm deep). Animals were head-plated, the DCN was exposed as described above, and a craniotomy above the VPL was performed. The dura was removed, and the area was flooded with 2% low-melting agarose dissolved in saline. A 32-channel multielectrode array (MEA; A1x32-poly2-10mm-50s-177-A32, Neuronexus) was lowered at 5 μm s$^{-1}$ into the brain. Once positioned in a region where firing in many units could be evoked by brushing the hindlimb, the MEA was kept in place for 20 min to ensure stable recording. The hindlimb was embedded in modelling clay for stabilization, and the receptive fields of units were quickly assessed using a brush. The mechanical stimulator probe tip was then placed over the region of the hindlimb that could maximally activate the most units. Step indentations of 0–300 mN were applied every 3–6 s in ascending order and repeated until at least 10 trials per force were obtained. Following stimulation, the DCN was lesioned either by using a 30-gauge needle and striking through the gracile nucleus or by aspirating the gracile nucleus. Step indentations were then repeated. Throughout the experiment, the waveform of a single unit was closely monitored to assess drift, and any experiment with detectable waveform changes were discarded. In some cases, once the experiment was complete, the MEA was retracted from the brain, coated with DiI and then descended to the same coordinates and allowed to stabilize for 5 min. Animals were then anaesthetized with isoflurane and transcardially perfused with PBS followed by 4% paraformaldehyde (PFA) to assess the location of the lesion and electrode placement.

For lesioning of the dorsal column, animals were prepared as described above, and a laminectomy was performed at approximately T12 vertebrae and the dura was removed. A high-density 32-channel MEA (A1x32-poly3–5mm-25s-177-A32, Neuronexus) was inserted into the gracile nucleus and allowed to stabilize for 15 min. A baseline of 5 min of spontaneous firing was collected. The dorsal column was then lesioned at approximately T12 using a 30 gauge needle. Brief vibratory stimuli were applied to the hindlimb throughout the experiment to assess the effectiveness of the lesion.

MEA recordings were made using an Intan head-stage, recording controller (Intan Technologies RHD2132 and recording controller) and open-source acquisition software (Intan Technologies RHX data acquisition software, v.3.0.4). Data were sampled at 20 kHz and bandpassed (0.1 Hz–7.5 kHz).

## Ex vivo recordings

Intracellular recordings were performed in random DCN neurons ex vivo. The brainstem was prepared as previously described[13], except DCN neurons were recorded directly from the dorsal surface of a dissected brainstem. Borosilicate electrodes (2–3 MΩ) filled with internal solution (consisting of in mM: 130 potassium-gluconate, 3 KCl, 10 HEPES, 0.5 EGTA, 3 MgATP, 0.5 NaGTP, 5 phosphocreatine-Tris$_2$ and 5 phosphocreatine-Na$_2$, pH 7.2 with KOH) were visually guided to the DCN. Experiments were performed in recirculated artificial cerebrospinal fluid (containing in mM: 127 NaCl, 2.5 KCl, 1.25 NaH$_2$PO$_4$, 1.5 CaCl$_2$, 1 MgCl$_2$, 26 NaHCO$_3$ and 25 glucose) oxygenated with 95% O$_2$/5% CO$_2$. The preparation was kept at 35 °C using an in-line heater. Recordings of random DCN cells were made using a Multiclamp 700B (Molecular Devices) and acquired using a Digidata 1550B with Clampex 11 software (Molecular Devices).

## Pharmacology

For spinal cord silencing experiments, animals were prepared for juxtacellular DCN recordings as described above. A laminectomy was performed over L3–L5 spinal segments, the region of the spinal cord responsive to hindlimb stimulation. For control experiments, a laminectomy was performed over T10–T13 spinal segments. The exposed spinal cord and vertebral column was then flooded with low-melting point agarose. Once hardened, agarose overlying the dorsal horn was cut away to create a pool and confine drugs to the spinal cord. The dura was removed from the spinal cord using fine forceps. Before drug application, stimuli were applied to the hindlimb to evoke typical responses to ensure that the spinal cord or dorsal column had not been damaged. Drugs were then applied to the spinal cord. The non-competitive NMDA receptor antagonist MK-801 (10 mM, 10 μl, Abcam; dissolved in 90% H$_2$O and 10% DMSO) was applied to the surface of the spinal cord and allowed to enter the cord for 3–5 min. The surface of the cord was then irrigated with saline. NBQX (10 mM, 20 μl, Abcam, dissolved in H$_2$O) was then applied to the surface of the cord. After 5 min, the cord was covered with gelfoam, which was allowed to absorb the NBQX and remained in place for the duration of the experiment, occasionally re-wet with saline. Drugs were sequentially applied to prevent precipitates from drug mixing at high concentration. Units were then recorded, and animals were sacrificed within 2–3 h following drug application.

For controls measuring the efficacy of blockade, a laminectomy was performed over the L4 spinal cord. The vertebral column was held in place by two custom spinal clamps. The dura was removed using fine forceps or a fine needle and kept moist with saline. A 32-channel MEA was inserted into medial L4 (A1x32-poly2–10mm-50s-177-A32, Neuronexus). Step indentations (300 mN) were applied at 0.1 Hz, and the MEA was lowered such that the most dorsal channel detected minimal evoked responses. The location of the dorsal channel was assumed to be near the surface of the cord. The MEA was then kept in place for 15 min. Baseline trials were collected and then drugs were applied as described above.

## Stereotaxic injections

Adult animals were anaesthetized with isoflurane and placed in a stereotaxic frame. The head was tilted 30° forward. The hair over the neck and caudal scalp were removed using a clipper and the skin was sanitized using isopropanol followed by betadine. Local anaesthesia (2% lidocaine HCl) was applied to the area, and an incision was made to expose neck muscles. Neck muscles were separated from the skull to expose the brainstem. A 30-gauge needle was used to cut the dura overlying the brainstem and to expose the DCN. A pipette filled with retrograde tracer (Red Retrobeads, Lumafluor or cholera toxin subunit B, 2 μg μl$^{-1}$, Fisher) and 0.01% fast green was lowered into the DCN just sufficient to penetrate the surface. Once penetrating the surface, a small volume (30–50 μl) of retrograde tracer was injected. The pipette was held in place for 20 s and then removed. Care was taken to ensure tracer did not leak from the injection site following injection and that tracer labelled with fast green filled the DCN but did not extend beyond the nucleus. The pipette was then removed, and overlying muscle and skin was sutured shut. Animals were administered analgesic (Buprenex SR, 0.1 mg kg$^{-1}$, ZooPharm) before surgery and monitored post-operatively. After 1–3 days, animals were transcardially perfused for tissue collection.

## Histology

Animals were anaesthetized with isoflurane and transcardially perfused with PBS followed by 4% PFA in PBS. Brains were removed and post-fixed in PFA overnight. The brain, spinal cord and dorsal root ganglia (DRGs) were dissected free. The isolated spinal cord or brain was mounted in low-melting point agarose, and sections of the thoracic spinal cord (60 μm thick) were made on a Leica VT1000S vibratome. For immunohistochemistry, free-floating sections were first permeabilized with 0.1% Triton-X100 in PBS for 30 min at room temperature. Sections were then incubated with 0.1% Triton-X100 and 4% normal goat serum (Abcam) for 30 min at room temperature. Primary antibodies were then added (mouse anti-NeuN, 1:1,000, MAB377 clone A60, Millipore; guinea-pig anti-VGLUT1, 1:2,000, Synaptic Systems 135302) and incubated overnight at 4 °C. Sections were then washed three times with PBS with 0.1% Triton-X100 and 4% normal goat serum for 10 min at room temperature. Secondary antibodies were then applied (IB4-Alexa 647, 1:300, ThermoFisher I32450; goat anti-mouse Alexa-488, 1:500, Abcam ab150113; FITC goat anti-GFP, 1:500, Abcam ab6662) for 2 h at room temperature and washed with PBS three times for 10 min. Sections were mounted onto glass slides using Fluoromount aqueous mounting medium (Sigma). Sections were imaged with a Zeiss LSM 700 confocal microscope using a ×20, 0.8 NA oil-immersion objective and Zen software. DRGs were whole-mounted and imaged using a Zeiss LSM 700 confocal microscope using a ×10, 0.45 NA air objective.

PSDC neurons were counted in z-stacks (3–4 μm z-spacing) of 60 μm thick thoracic spinal cord sections. A total of 10–20 sections were analysed per animal, and the average number of PSDC neurons per section was measured and multiplied by 16.67 to estimate the number PSDC neurons in one segment (1,000 μm) of spinal cord for that animal. Aβ-LTMRs were counted in whole-mounted thoracic DRGs and compared to spinal cords of the same animals. Labelled DRG neurons were counted in z-stacks of the entire DRG (5–6 μm z-spacing). If more than one DRG was analysed per animal, the number of counted cells was averaged between the two.

## Behaviour

Behaviour experiments were performed in two C57Bl/J6 males and four *Calca-FlpE;Rosa26*$^{FSF-ReaChR}$ animals (2 male, 2 female). Animals were anaesthetized with 2% isoflurane. Hair over the scalp was clipped using a shaver and skin was disinfected using isopropanol followed by betadine. Local anaesthesia (2% lidocaine HCl) was applied to the scalp. An incision was made and the dorsal skull was cleared of skin

and muscle using a scalpel. A head plate was rested on top of the skull and fixed in place using dental cement. Animals were administered analgesic (Buprenex SR, 0.1 mg kg$^{-1}$, ZooPharm) before surgery and monitored post-operatively.

Animals were allowed to recover for 2–3 weeks. Head-plated animals were then transferred to a rig consisting of an acrylic platform. Animals were head-fixed to a suspended post, but were otherwise free to move. Mice were allowed to habituate to the rig for 5–10 min. Gentle mechanical stimuli were then delivered to prevent startle responses. First a brush was used to gently stroke the trunk and hindlimbs. After 5–10 strokes with a brush, the indenter was introduced and 300 ms indentations of 10 mN were delivered using the same smoothed 1 mm probe tip used for the electrophysiology experiments. The indenter and probe tip were the same as those used for experiments in anaesthetized animals. Indentations were delivered to the middle of the dorsal hindlimb as the ventral side was inaccessible in awake animals with unrestrained paws. When the animal was still, the indenter tip was positioned in place above the hindlimb, a trial was triggered and indentation was delivered after 1 s of baseline. Stimuli were delivered approximately every 20 s. After a few trials using 10 mN stimuli, animals no longer were startled by indentation, and the force was increased to 300 mN. Trials were recorded using a CCD camera (BlackFly S BFS-U3-04S2C, Flir) attached to a stereoscope with frames captured at 100 Hz using Spinview software. The first trial of 300 mN typically evoked a startle response and was therefore discarded. The subsequent trials were then collected for analysis (5–12 trials). Several seconds following 300 mN indentation, a small force (<10 mN) was briefly applied to the probe to bring it into position near the skin again for the subsequent trials.

In four *Calca-FlpE;Rosa26*$^{FSF-ReaChR}$ animals, once mechanical stimulation was completed, a set of trials were performed in which the skin was optically stimulated. A 400 μm 0.39 NA fibre coupled to a 554 nm LED was held in place above the centre of the dorsal hindlimb. A 50 ms pulse of light (about 30 mW mm$^{-2}$) was delivered to the skin. The first trial was discarded, and subsequently 4–7 trials were collected for analysis. Experiments were ended after fewer trials because animals showed clear signs of pain.

To determine whether mechanical stimulation generated enough force to prevent the animals from withdrawing their paw, optical stimulation was delivered during mechanical stimulation in two *Calca-FlpE; Rosa26*$^{FSF-ReaChR}$ animals. A 300 mN, 300 ms indentation was delivered as described above. At 100 ms after the onset of indentation, a 50 ms pulse of light was delivered at or near the indentation as described above. Animals readily withdrew their hindlimb within 30 ms of light onset when 300 mN of force was still being applied.

## Analysis

Juxtacellular recordings in the DCN were analysed offline using custom written scripts in Matlab (Mathworks). Spikes were detected using an amplitude threshold, and recordings in which the unit could not be isolated by amplitude were discarded.

Receptive field maps were generated using custom written scripts in Matlab. An image of the hindlimb from the experiment was used as a template. Images capturing the probe or optic fibre for each trial were cycled through, and the location of the probe or fibre tip was manually marked on the template image. The coordinates of each stimulus location and the template image were then combined with electrophysiology data to identify the number of evoked spikes for each stimulus location. The receptive field map for each unit is displayed with points indicating the location of the stimulus and normalized change in firing rate from baseline. The maximal change in firing rate is indicated in text within the image. The maximal firing rate was measured as the number of spikes over the course of the 100 ms vibratory stimulus or the number of spikes within a 20 ms window following optical stimulation. The spontaneous firing rate was measured as the average firing rate before mechanical stimulation across trials. To measure the receptive

field area, a threshold was taken at the half-maximal evoked firing rate. Stimulus locations that evoked the half-maximal firing rate or more were included in the receptive field. The area bound by these points was used as a measure of the receptive field area.

Correlations of optical and mechanical receptive fields were determined by comparing regions of the hindlimb. First, the optical and mechanical receptive fields were mapped as described above. The paw was then subdivided into segments corresponding to digits and pads. The stimulus trials falling within each segment were averaged to obtain an average evoked firing rate for each segment. The optically evoked firing rate for each segment was plotted against its mechanically evoked firing rate for that same region. This generated a plot of 17 points, each representing the optical and mechanical response of one region of the hindlimb. A linear fit was performed, and the Pearson's correlation coefficient ($R^2$) was used as a measure of receptive field correlation. As a control, a similar analysis was performed but optical and mechanical receptive fields were shuffled between units.

### Spike sorting (MEAs)

MEA recordings underwent initial analysis using Kilosort 2.0 (ref. [53]). Before analysis, baseline and lesions trials were interleaved to prevent artefactual drift corrections by Kilosort. Drift monitoring was performed during acquisition, and experiments with detectable changes in spike waveforms were discarded. Default detection settings were used for analysis, except that template amplitude thresholds were set to [5 2]. Clusters were manually curated using Phy[54] by examining spike waveforms and autocorrelations to identify putative single units. As receptive fields were not mapped for MEA experiments, units underwent selection for inclusion. Units that had low thresholds at baseline (<10 mN) or units that had a mechanically evoked firing that was unaffected by lesioning the DCN were included. Units that had high thresholds (>10 mN) at baseline and became mechanically insensitive following DCN lesion were not included for analysis because the stimulus may have been off the centre of the receptive field.

$Z$-scores were computed by measuring the average baseline firing rate within a 0.5–1 s window before stimulation. For optical silencing, pharmacology and lesion experiments in which the same units were monitored before and after manipulation, the $Z$-scores in experimental conditions were computed from the mean and standard deviations of baseline trials.

### Behaviour

Videos of mechanical and optical stimuli delivered to the hindlimb were collected in trials lasting 3 s. The paw position was semiautomatically tracked using the Video Labeler application in Matlab. Baseline paw position was collected 0.5 s before stimulus delivery for the trial. The baseline position was set as the origin, and the position of the paw relative to the starting position was measured for the entire trial. For average plots, the position was averaged across trials for each animal. For total movement, the total change in position was summed over the average trial for each animal.

### Statistical analyses

Statistical analyses were performed with the significance threshold set at $P < 0.05$. All summary data are presented as the mean ± s.e.m. unless otherwise noted. All $t$-tests were two-tailed.

### Reporting summary

Further information on research design is available in the Nature Portfolio Reporting Summary linked to this article.

### Data availability

Datasets may be obtained from the corresponding author upon reasonable request. Source data are provided with this paper.

### Data availability

Custom scripts used in this study are posted on Github (https://github.com/josefturecek/Turecek_Nature_2022) with additional information available upon request from the corresponding author.

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

**Acknowledgements** We thank C. Harvey, C. Chen, W. Regehr, C. Santiago, A. Emanuel, G. Rankin, A. Shuster, A. Chirila, R. Martinez-Garcia, D. Zhang and J. Barowski for comments on the manuscript. This work was supported by a Mahoney Postdoctoral Fellowship (to J.T.), a Gordon Postdoctoral Fellowship (to J.T.), NIH grants NS097344 and AT011447 (to D.D.G.), The Hock E. Tan and Lisa Yang Center for Autism Research at Harvard University (to D.D.G.), and the Edward R. and Anne G. Lefler Center for Neurodegenerative Disorders (to D.D.G.). D.D.G. is an investigator of the Howard Hughes Medical Institute. This article is subject to HHMI's Open Access to Publications policy. Heads of HHMI laboratories have previously granted a nonexclusive CC BY 4.0 licence to the public and a sublicensable licence to HHMI in their research articles. Pursuant to those licences, the author-accepted manuscript of this article can be made freely available under a CC BY 4.0 licence immediately upon publication.

**Author contributions** J.T., B.P.L. and D.D.G. conceived the project. J.T. performed all experiments except data shown in Extended Data Fig. 3. J.T. performed all analyses. B.P.L. performed pilot experiments not presented and anatomical experiments shown in Extended Data Fig. 3. J.T. and D.D.G. wrote the paper with input from B.P.L.

**Competing interests** The authors declare no competing interests.

**Additional information**
**Correspondence and requests for materials** should be addressed to David D. Ginty.

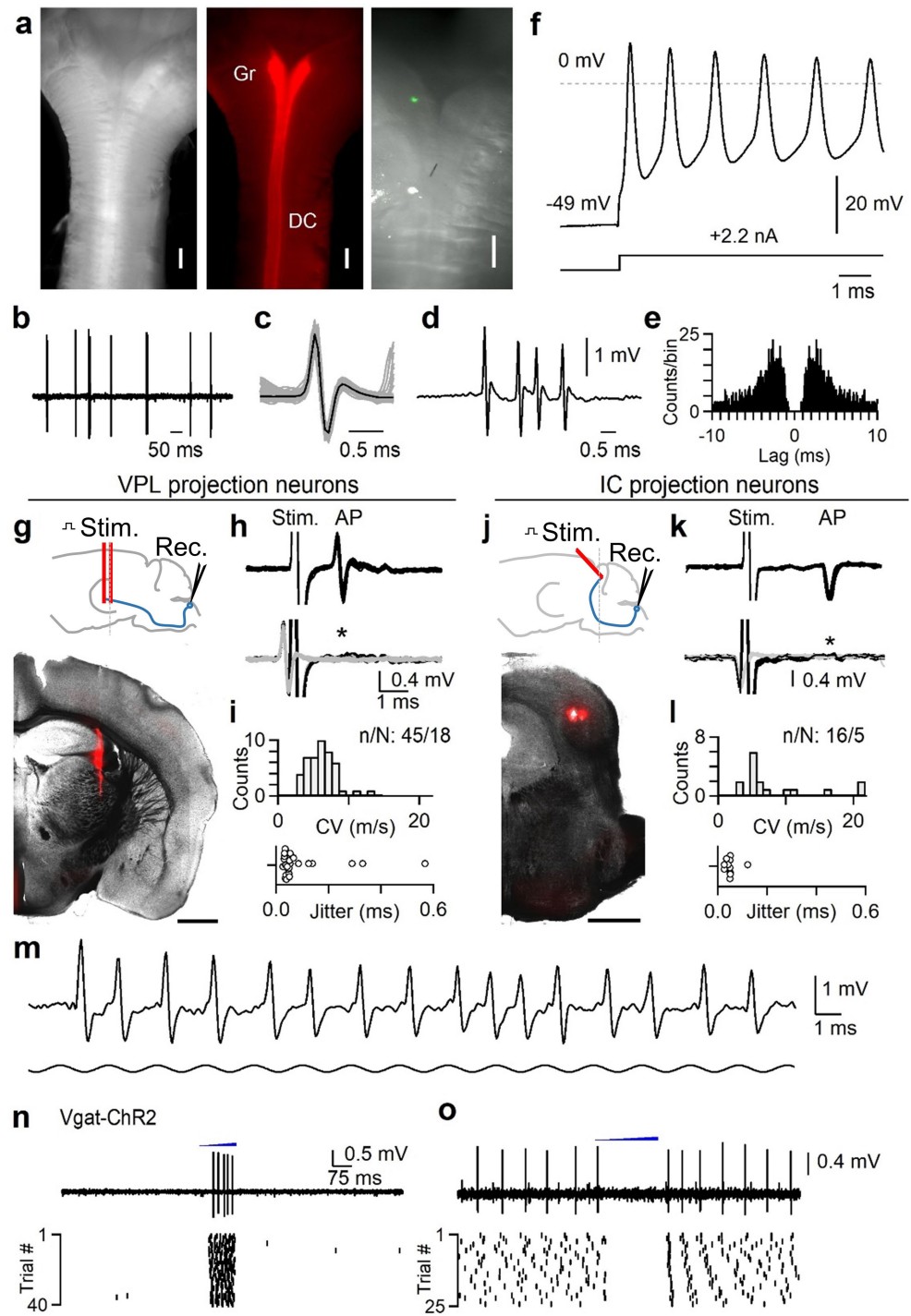

**Extended Data Fig. 1** | See next page for caption.

**Extended Data Fig. 1 | Juxtacellular recording in the DCN, antidromic activation, and optotagging.** Juxtacellular recordings were made from units in the gracile nucleus of the DCN to obtain high signal-to-noise recordings of putative single units. Recordings were obtained in urethane-anesthetized mice. Units in the DCN can fire at very high rates up to 1 kHz and can undergo use-dependent changes in action potential waveforms during bursts[55]. Many response properties seen in non-anesthetized preparations are present in anesthetized animals, including a wide distribution of receptive field sizes, entrainment to high-frequency vibration, inhibitory surround receptive fields, and persistent responses to sustained stimuli[39–41]. However, urethane anesthesia will suppress descending input that may modulate the DCN, and may alter polysynaptic circuits that affect DCN activity during different behavioral states. **a**, _Left_: Top view showing the dorsal surface of an isolated brainstem with myelinated dorsal columns forming a "Y" shape as they reach the DCN. _Middle_: AAV1-hSyn-Cre was injected into the lumbar cord of a mouse expressing _Rosa26_[LSL-Acr1-FRed], labeling the dorsal columns (DC) and gracile nucleus (Gr). _Right_: Zoomed-in view of the gracile nucleus following a recording when the electrode was coated with DiI. The location of the recording site can be seen in green. **b**, Example of spontaneous firing recorded juxtacellularly in the DCN _in vivo_. **c**, Overlaid spikes with single events (gray) and average waveform (black). **d**, Example high-frequency burst. **e**, Autocorrelogram for unit in **a**–**d**. **f**, _Ex vivo_ whole-cell recording of a DCN neuron from an intact brainstem of an adult mouse. Recording was performed in near-physiological conditions (35 °C). A current injection was supplied to trigger a brief burst of very high frequency firing. Cells could fire up to 1 kHz bursts with use-dependent changes in action potential waveform as observed _in vivo_. Three random large-diameter cells were recorded _ex vivo_ in one animal. **g**, Identification of VPL-projection neurons (VPL-PNs). _Top_: schematic of preparation, a bipolar electrode was lowered into the VPL. _Bottom_: Example coronal section (100 μm thickness) showing track of the stimulating electrode that was coated with DiI. Scale bar is 1 mm. Anatomical location of stimulus electrode was verified in two animals. **h**, _Top_: A unit in the DCN antidromically activated from the VPL. Overlay of 20 trials shows consistent waveform of the stimulus artifact (stim.) and antidromic action potential (AP). The conduction velocity was calculated from the latency to spike following the stimulus artifact with an estimated distance of 9.8 mm. _Bottom_: during a collision test, the stimulus was triggered from the spontaneous firing of the unit. Overlay of 20 trials when electrical stimuli were triggered from spontaneous firing (black). In these trials, stimulation failed to evoke an antidromic spike that would be expected at the latency indicated by the asterisk. Waveform of spontaneous firing in which stimuli were not triggered are overlaid in gray. **i**, _Top_: Histogram of calculated conduction velocities for units antidromically activated from the VPL. Distance from the DCN to VPL was estimated to be 9.8 mm. _Bottom_: Jitter (standard deviation) of latency from units antidromically activated from the VPL that passed collision testing. **j**, Identification of IC-projection neurons (IC-PNs). _Top_: schematic of preparation, a bipolar electrode was angled at 45° in order to avoid the sinus above the IC. _Bottom_: Example coronal section (100 μm thickness) showing track of the stimulus electrode that was coated with DiI. Scale bar is 1 mm. Anatomical location of stimulus electrode was verified in two animals. **k**, Same as in **h**, but for a unit antidromically activated from the IC. **l**, Same as in **i**, but for units activated from the IC. Distance from the DCN to IC was estimated to be 9.4 mm. IC units are those that were classified as vibration-tuned only (see Extended Data Fig. 2). **m**, Juxtacellular recording of a single DCN unit _in vivo_ showing response to high frequency (500 Hz) vibration of the hindlimb. The unit could entrain to single cycles of the vibration and underwent use-dependent changes in waveform. Vibration begins 6 ms prior to displayed time window. **n**, Optotagging of inhibitory interneurons in the DCN in a Vgat-ChR2 mouse. A fiber optic was positioned above the DCN (400 μm diameter, 1 mm away). Light from a fiber-coupled LED was ramped for 200–300 ms (470 nm, 1-2 mW/mm$^2$). Optotagged units fired during the ramp. A single trial (_top_) and raster of all trials (_bottom_) for an optotagged unit are shown. **o**, Units that were not optotagged were suppressed by light ramps, or fired rebounds after termination of the ramp. Data shown as n units in N animals (n/N).

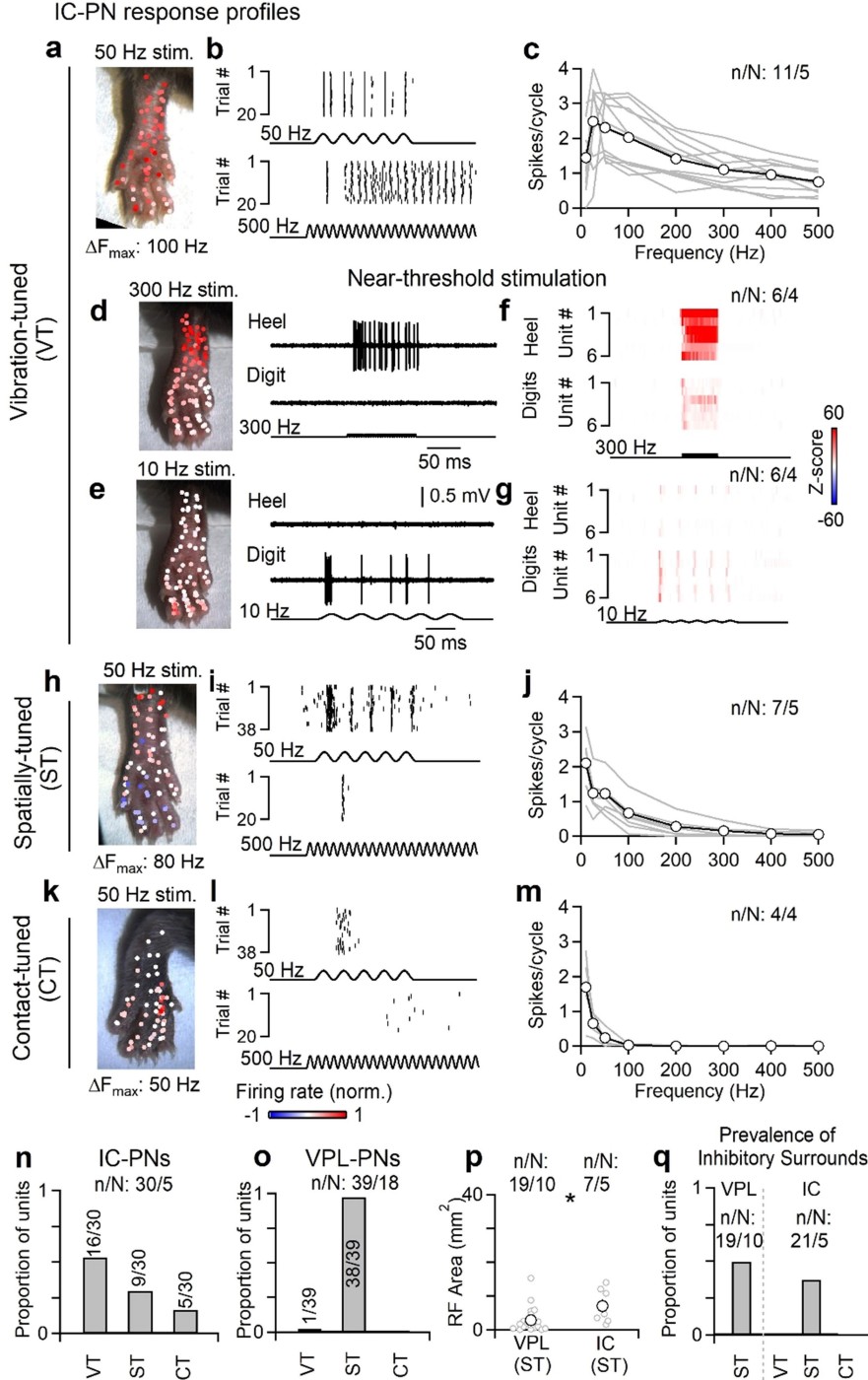

**Extended Data Fig. 2** | See next page for caption.

**Extended Data Fig. 2 | IC projection neuron subtypes.** The DCN is composed of several types of projection neurons and local interneurons. VPL-PNs are the most abundant, with an estimated proportion of VPL-PN:IC-PN:Vgat-IN of 2:1:1 based on retrograde tracing and genetic labeling in mice[13]. The proportion of VPL-PNs may be larger[12]. Units that could be antidromically activated from the IC fell into three functional types with distinct properties. Vibration-tuned (VT) units have very large receptive fields that include almost the entire hindlimb, they can entrain their firing to very high frequencies (>300 Hz and often up to 500 Hz), and do not have inhibitory surrounds. The broad receptive field and entrainment to high frequencies suggest that they receive strong input from Pacinian corpuscles. Spatially-tuned (ST) units have small receptive fields, typically consisting of 1-2 digits or pads, or a small portion of the heel. They often have inhibitory surrounds, and can entrain their firing to modest vibration frequencies (up to 100–200 Hz). These units cannot entrain their firing to vibration frequencies > 300 Hz, even at high forces. Contact-tuned (CT) units have receptive field sizes similar to spatially-tuned units, but are poorly activated by vibration. They are very sensitive and almost exclusively responsive to the initial contact of a stimulus. **a**, Example receptive field of a vibration-tuned IC-PN. Color scale of normalized firing rate shown at bottom. **b**, Example rasters of a vibration-tuned IC-PN unit in response to 50 (*top*) or 500 Hz (*bottom*) vibration. **c**, Responsiveness of vibration-tuned IC-PN units to different vibration frequencies (10–20 mN). Data is shown for individual units (gray) and average across units (black). **d**–**g**, Vibration-tuned units had different sensitivities to different frequencies of vibration. When reducing the force of vibrations to near-threshold (<10 mN), 300 Hz vibrations evoked more robust responses in the heel, whereas 10 Hz vibrations typically only evoked responses when stimulating the digits. **d**, Receptive field map of near-threshold 300 Hz vibration for an identified vibration-tuned IC-PN (*left*). Example trials from the heel (*top right*) and digits (*top left*) are shown. Color scale of normalized firing rate shown at bottom. **e**, Same as in **d**, but for 10 Hz near-threshold vibration. Color scale of normalized firing rate shown at bottom. **f**, Average histograms for individual units of near-threshold 300 Hz vibration delivered to the heel (*top*) or digits (*bottom*). Color scale of Z-scores shown at right. **g**, Average histograms for individual units of near-threshold 10 Hz vibration delivered to the heel (top) or digits (bottom). Color scale of Z-scores shown at right. **h**–**j**, Same as **a-c**, but for spatially-tuned units. **k**–**m**, Same as **a**–**c**, but for contact-tuned units. **n**, Distribution of functional types for units antidromically activated from the IC. **o**, Functional types were generalized to classify VPL-PNs. Units antidromically activated from the VPL were almost all spatially-tuned. Their receptive fields were small and did not phase-lock to high frequency vibration (>300 Hz), even when increasing the amplitude. These units also typically did not elevate their firing in response to high-frequency vibration. 1/39 units were found to be vibration-tuned. **p**, Receptive field area for spatially-tuned units antidromically activated from the IC or VPL. Distributions are significantly different (p = 0.031, Kolmogorov-Smirnov test). Individual units (gray) and mean ± s.e.m. (black). **q**, Proportion of units with inhibitory surrounds for each functional type. All data shown as n units in N animals (n/N).

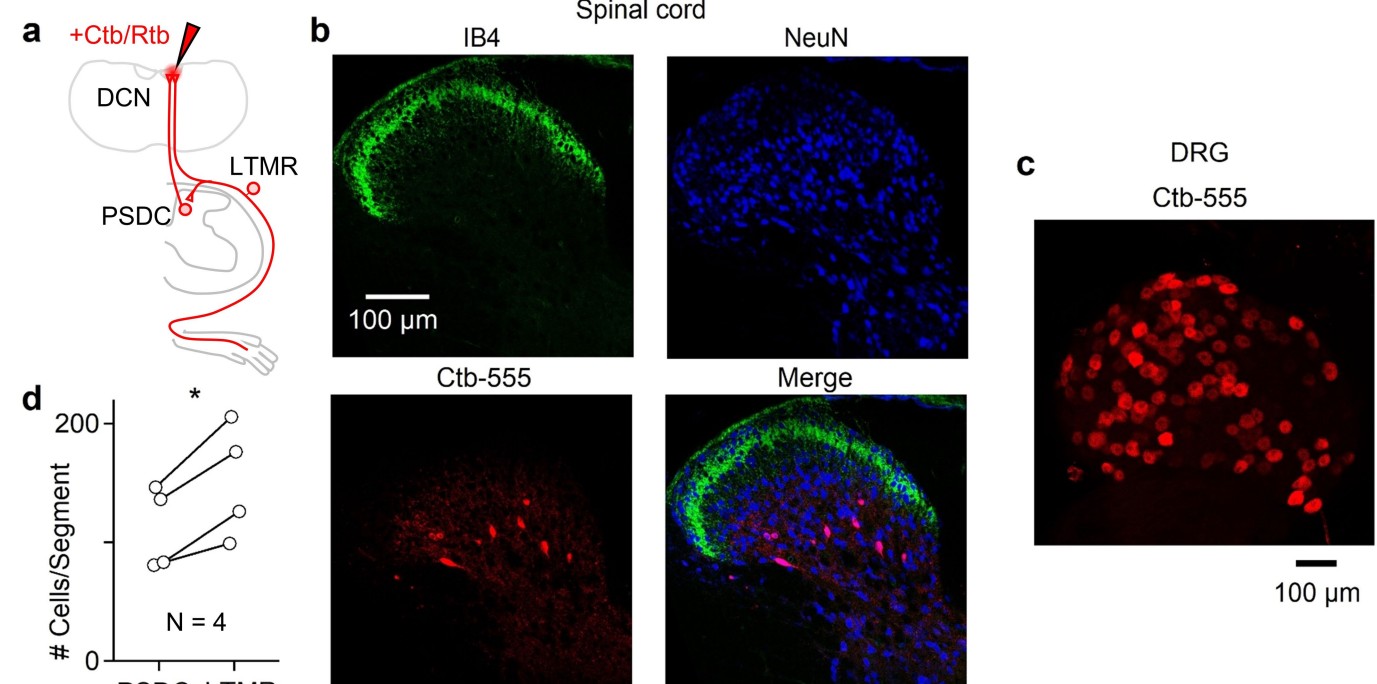

**Extended Data Fig. 3 | Prevalence and anatomical location of PSDCs.**
Choleratoxin-B (Ctb) or Retrobeads (Rtb) were injected into the DCN or C1 DC to label PSDCs and Aβ-LTMRs that project to the DCN. Two different tracers were used in order to avoid potential differences in LTMR vs. PSDC uptake efficiency. Small injections were performed in order to avoid off-target labeling in the spinal cord, and to compare the relative number of thoracic LTMRs and PSDCs. The number of LTMRs and PSDCs per thoracic spinal segment within the same animal was compared. PSDCs were found in lamina III-IV as described by others[6,31,56]. Experiment was performed in 4 animals. (N = 4 animals). **a**, Schematic showing injection of retrograde tracer into the brainstem and uptake by either PSDCs in the dorsal horn of the spinal cord or Aβ-LTMRs in the DRG. **b**, 10 μm Z-projection of a transverse thoracic spinal cord section (60 μm thick) with immunofluorescence of IB4 (*top left*, green), NeuN (*top right*, blue), Ctb-555 (*bottom left*), and merged images (*bottom right*). Experiment was repeated in 4 animals (see **d** for summary). **c**, Single plane image of a whole-mount thoracic DRG containing sensory neurons labeled with Ctb-555 from the same animal shown in **b**. Experiment was repeated in 4 animals (see **d** for summary). **d**, Comparison of number of Ctb or Rtb labeled cells in the thoracic spinal cord. Each marker pair is the average from one animal, showing the average number of cells labeled in 1-2 DRGs, and the number of PSDCs estimated to exist in one spinal segment (1000 μm length of spinal cord) based on the average number of PSDCs labeled in 10–20 spinal sections (60 μm thickness). p = 0.017, paired t-test.

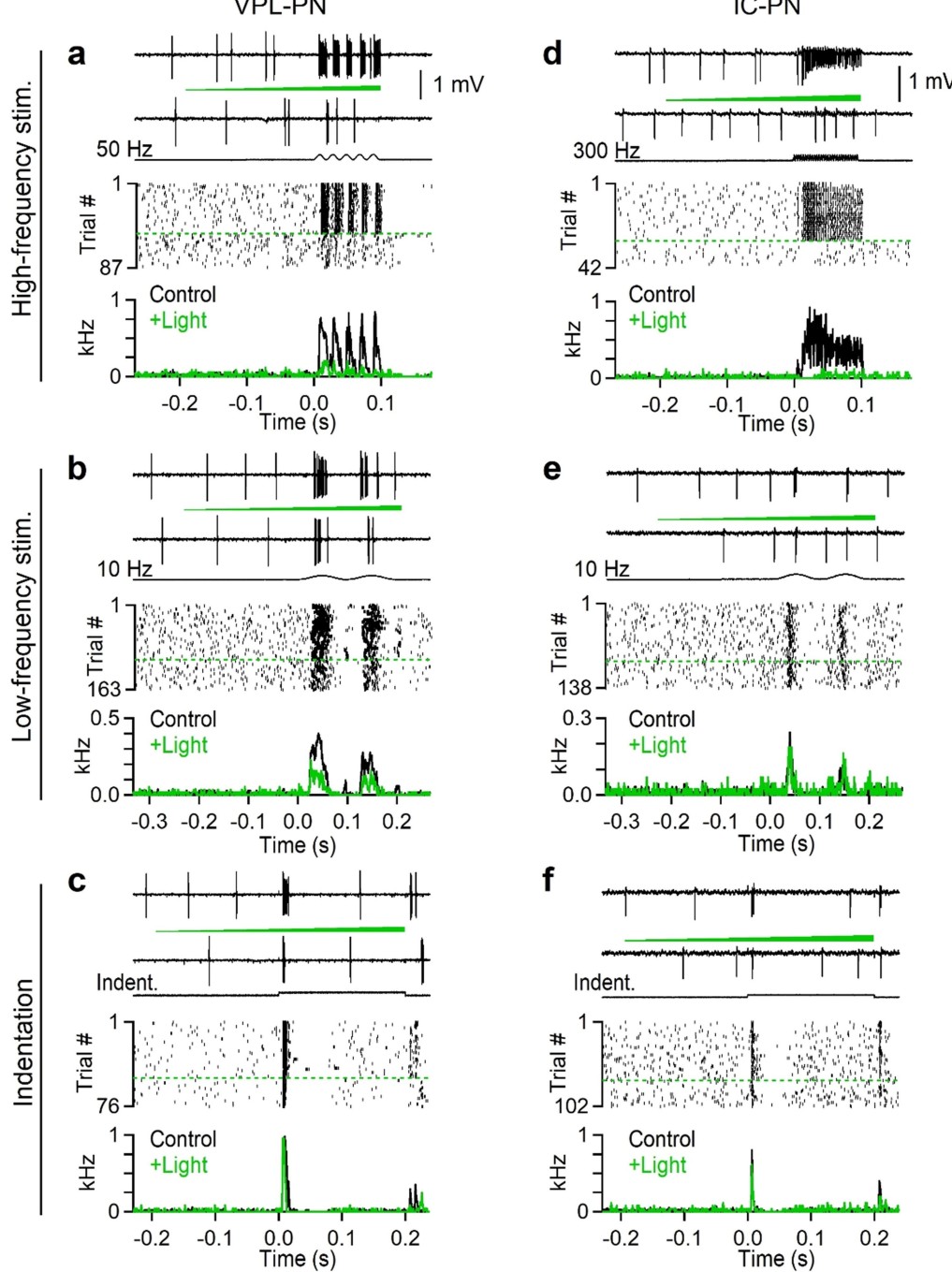

**Extended Data Fig. 4 | Effects of silencing sensory neuron terminals in the DCN using Acr1 on the response properties of identified projection neurons.** Juxtacellular recording of example VPL-PN (left column) and a vibration-tuned IC-PN (right column) in *Avil^Cre^;Rosa26^LSL-Acr1^* animals. In this experiment direct LTMR input is silenced. Mechanical stimuli were applied to the center of the receptive field. On some trials, light (550 nm, 0.46 mW/mm²) was applied to the surface of the brainstem using a fiber optic (400 μm diameter, 1 mm away). The light was ramped over the course of 300–400 ms, and mechanical stimuli were delivered in the last 100–200 ms of light. Trials were interleaved but are sorted here for presentation. One identified VPL-PN unit and one identified IC-PN was collected from two different animals. **a–c**, A DCN unit antidromically activated from the VPL. This unit did not fire in response to high-frequency (≥300 Hz) vibration, even at high forces. **a**, *Top*: Traces of single control and silencing trials. A 100 ms, 50 Hz, 10 mN vibration

was applied. Timing of light delivery is denoted by a green ramp. *Middle*: Raster of trials for 50 Hz vibration. Control and light trials are separated by a dashed green line. *Bottom*: Average histogram for this unit when light was off (Control) or on (+Light). Time 0 is the onset of vibration. Bins are 1 ms. **b**, Same unit as in **a**, but for a 200 ms 10 Hz 10 mN vibration. **c**, Same unit as in **a** and **b**, but for a 200 ms 10 mN indentation. **d-f**, A DCN unit that was antidromically activated from the IC. This unit entrained to 300 Hz vibratory stimuli and had a large receptive field. **d**, *Top*: Traces of single control and silencing trials. A 100 ms, 300 Hz, 10 mN vibration was applied. *Middle*: Raster of trials for 300 Hz vibration. Control and light trials are separated by a dashed green line. *Bottom*: Average histogram for this unit when light was off (Control) or on (+Light). Time 0 is the onset of vibration. Bins are 1 ms. **e**, Same unit as in **d**, but for a 200 ms 10 Hz 10 mN vibration. **f**, Same unit as in **d** and **e**, but for a 200 ms 10 mN indentation.

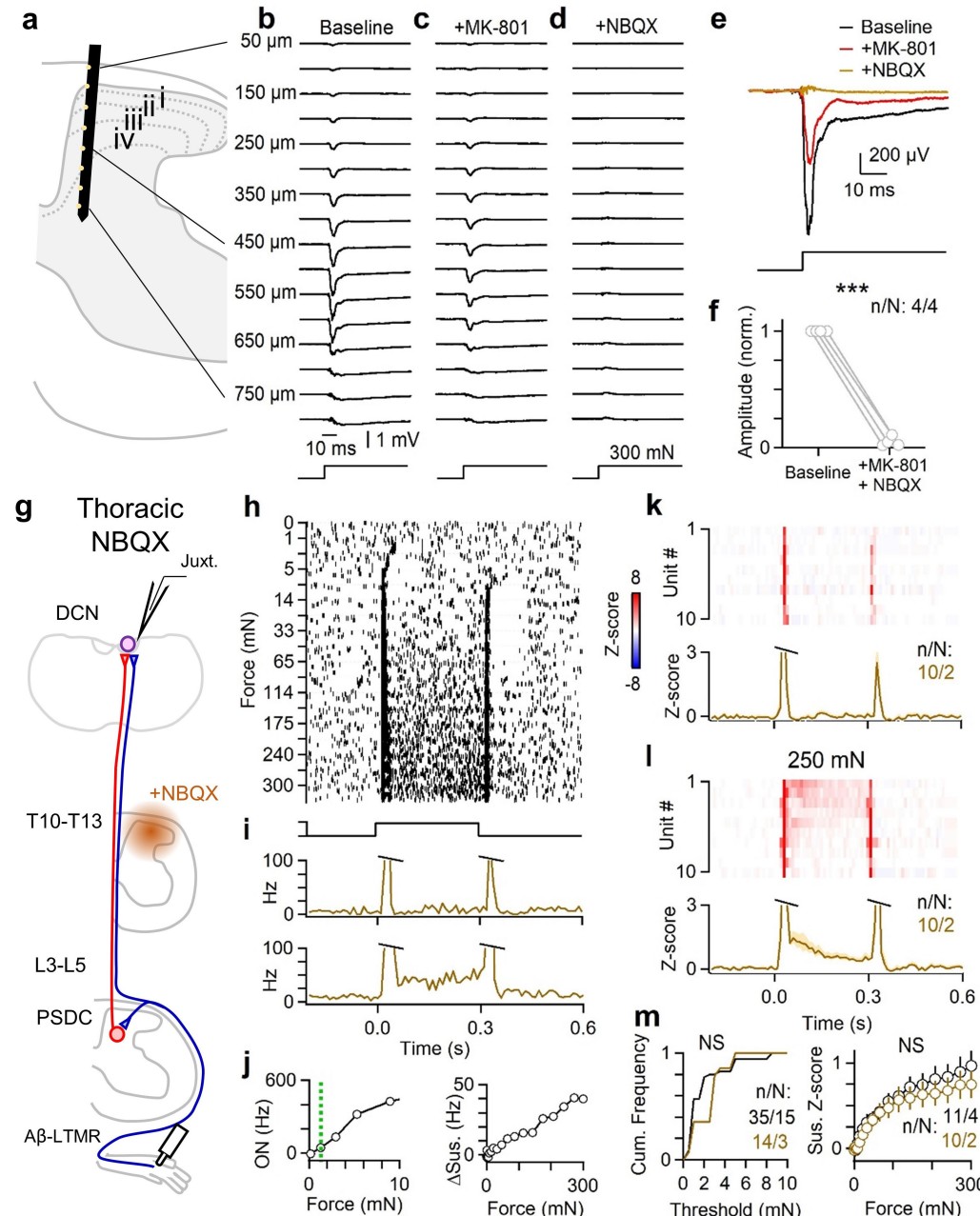

**Extended Data Fig. 5** | See next page for caption.

**Extended Data Fig. 5 | Blockade of glutamatergic transmission suppresses evoked responses locally within the spinal cord. a**–**f**, A multi-electrode array (MEA) was inserted into L4 spinal dorsal horn, with recording sites facing medially. Field potentials were recorded in response to a 300 mN step applied to the center of the receptive field on the hindlimb. Following baseline trials, MK-801 (10 mM, 10 μL) was applied to the surface of the spinal cord for 3–5 min to block NMDA receptors. The surface was then irrigated with saline and NBQX (10 mM, 20 μL) was applied to block AMPA receptors. **a**, Schematic of approximate recording location. MEA has 16x2 recording sites spaced 50 μm apart. Only a subset of sites are schematized. **b**, Field potentials generated at various depths of the spinal cord in response to a 300 mN step indentation under baseline conditions. **c**, Same recording as in **b**, but after application of MK-801 to the surface of the spinal cord. **d**, Same recording as in **b** and **c**, but after application of NBQX to the surface of the spinal cord. **e**, Overlaid field potentials from a 450 μm depth recording site in response to 300 mN indentation, from recording shown in **b**–**d**. Responses following NBQX remained suppressed for the remainder of the recording (30 min after NBQX application). **f**, Summary of change in evoked amplitude. Each marker pair is one animal. (Paired t-test, p < 0.001). **g**–**m**, Pharmacological manipulations can have non-specific effects, as drugs may spread from the site of application or enter the body systemically. We assessed these possibilities by applying glutamatergic antagonists to the thoracic cord (T10–T13) to determine whether hindlimb responses in the DCN may be altered through diffusion of the drug to the DCN or non-targeted regions of the spinal cord. A laminectomy was performed over the T10–T13 spinal segments, and the dura was removed. MK-801 (10 mM, 10 μL) was first applied to the surface of the spinal cord for 5 min. The cord was then abundantly irrigated with saline. NBQX (10 mM, 20 μL) was then applied to the surface and gelfoam was placed on top and kept moist with saline. **g**, Schematic of experimental setup. **h**, Raster of example DCN unit in response to step indentations of various forces (0–300 mN) in the presence of NBQX applied to T10–T13 spinal segments. **i**, Histogram of responses to 10 mN (*top*) and 250 mN (*bottom*) for unit shown in **h**-**i**. **j**, *Left*: Max on-response (0–20 ms) vs. indentation force. Dashed green line indicates threshold. *Right*: Average sustained firing (50–250 ms) vs. force. **k**, *Top*: Response of DCN units to 10 mN step indentation in the presence of NBQX. *Bottom*: Mean ± s.e.m. across DCN units to 10 mN indentation in the presence of NBQX. **l**, Same as K, but for 250 mN step indentation. **m**, *Left*: Mechanical threshold of on-response for units in the presence (brown) or absence (black) of NBQX over T10–T13 spinal segments. Distributions are not significantly different (Kolmogorov-Smirnov test, p = 0.25). *Right*: Mean ± s.e.m. of sustained firing rate as a function of indentation force across cells in the presence (brown) or absence (black) of NBQX over T10–T13. (Kolmogorov-Smirnov test, p = 0.76). Data shown as n units in N animals (n/N).

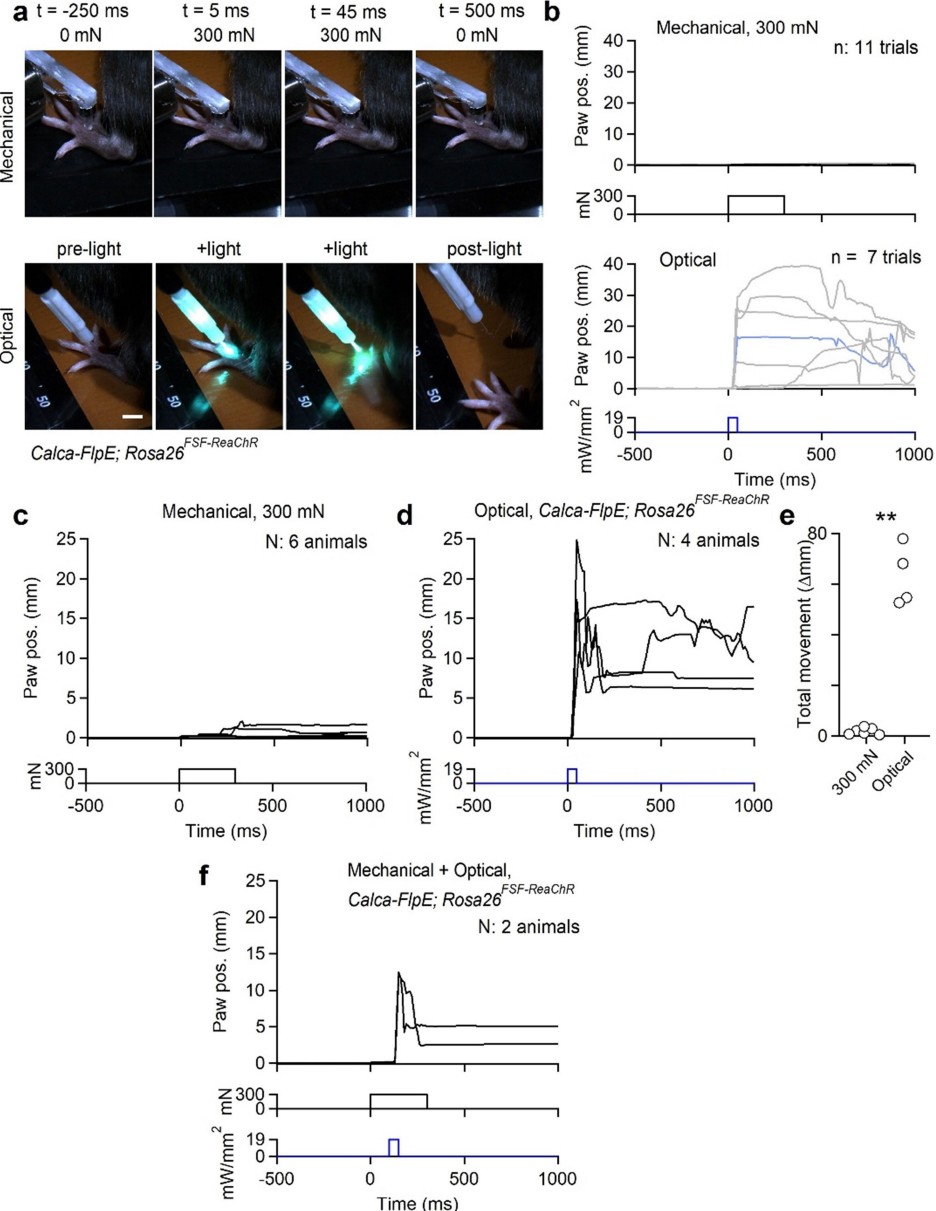

**Extended Data Fig. 6 | Mechanical stimuli used for physiology experiments do not evoke nocifensive responses.** Indentation at high forces (300 mN) using a large, 1 mm probe tip was not perceived as painful when applied to the finger tip of human experimenters (J.T. and D.D.G.). We tested whether this same high force stimulus was noxious for mice. Awake C57Bl/J6 or *Calca-FlpE*; *Rosa26^FSF-ReaChR* animals were head-fixed and otherwise unrestrained. Stimuli were delivered to the paw and the mouse was free to withdraw its paw or attempt to avoid the stimulus. Animals were allowed to habituate without stimuli for 5–10 min. Then, gentle mechanical stimuli (brush) were delivered to prevent startle responses. Weak (10–20 mN) indentations were then delivered to the hindpaw to acclimate the animal to the 1 mm probe. Then the amplitude was elevated to 300 mN and data collection began. The first 300 mN indentation trial was discarded to avoid capturing startles. 4–15 trials were then collected to determine whether the mouse withdrew its paw. In *Calca-FlpE;Rosa26^FSF-ReaChR* animals, the series of indentation trials were then followed by a series of optical stimulation trials. A 400 μm fiber delivered a ~20 mW/mm² 50 ms pulse of light to the dorsal hindpaw. The first trial was discarded, and 3–7 trials were then collected. See also Supplemental Video 1. **a**, Video frames acquired during either indentation (top) or optical activation of *Calca⁺* afferents (bottom) in the same mouse. Scale bar in bottom left panel is 5 mm. **b**, Plot of paw displacement (paw position) relative to baseline for 300 mN 300 ms indentation for the mouse shown in **a** (top), and plot of paw displacement relative to baseline for 50 ms optical stimulation for the same experiment shown in **a** (bottom). Each trace is one trial from the same animal. Highlighted blue trace is from trial shown in **a**. **c**, Average paw displacements (paw position) for 6 mice in response to 300 mN indentation. Each trace is the average of one animal. **d**, Average paw displacements (paw position) for 4 mice in response to optical stimulation of *Calca⁺* afferents. **e**, Total average movement (distance paw traveled) during mechanical or optical stimulation. Each marker is the average of one animal. Kolmogorov-Smirnov test, p = 0.005. **f**, Lack of paw withdrawal following 300 mN mechanical stimulation was not due to constraint of the paw by the indenter. In two *Calca-FlpE;Rosa26^FSF-ReaChR* animals, a 300 mN indentation was delivered. 100 ms after the onset of indentation, light was also delivered to the paw. Optical stimulation triggered paw withdrawal in the presence of a 300 mN force pressing down on the paw. Data shown as n units in N animals (n/N).

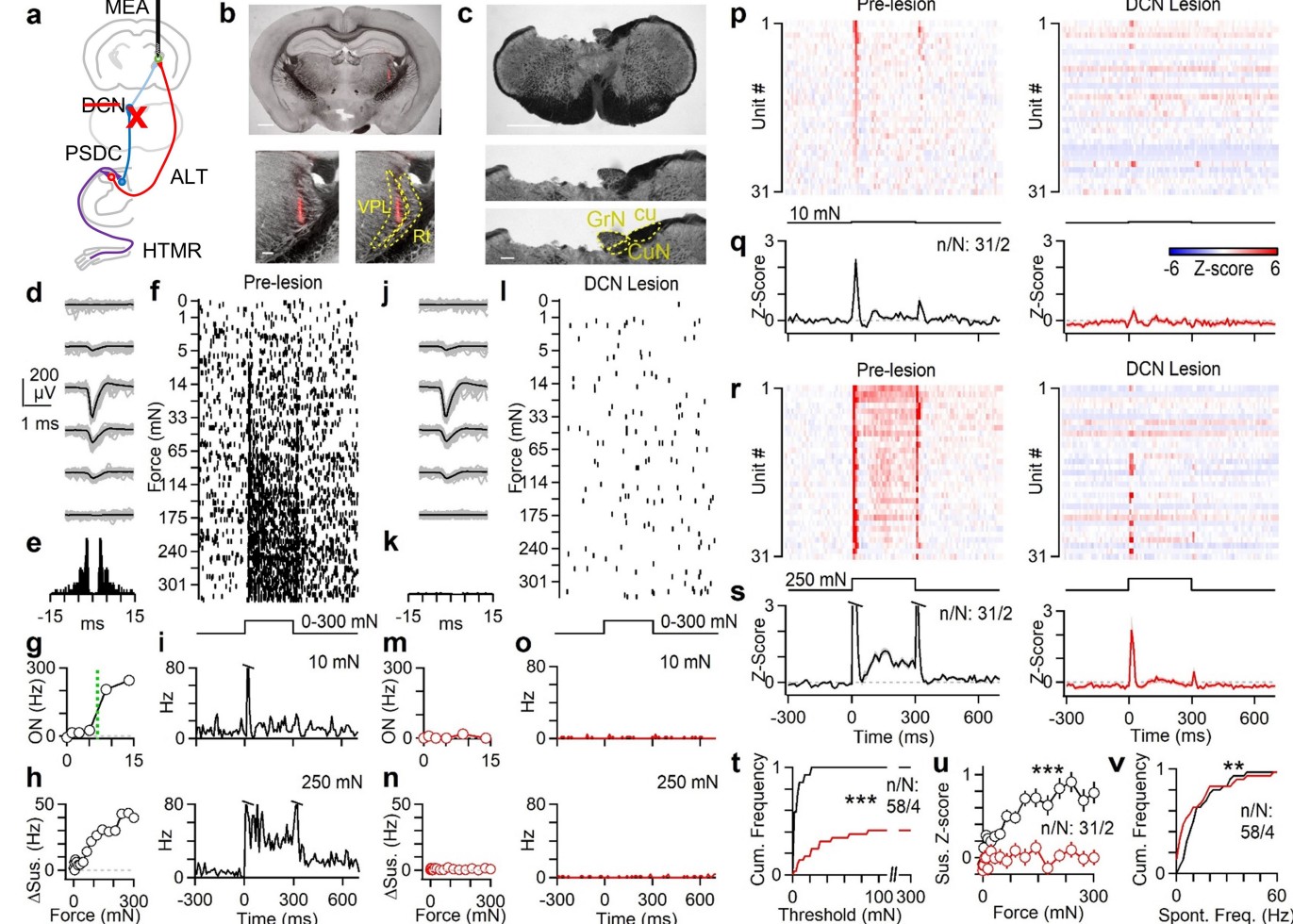

**Extended Data Fig. 7 | The DCN supplies high force information to the middle VPL (mVPL).** The primary somatosensory thalamus can be subdivided into rostral, middle and caudal regions[57]. Rostral regions may receive more proprioceptive input and have large receptive fields. The middle region has cutaneous responses with detailed spatial receptive fields. The caudal region has diffuse receptive fields and may be more responsive to intense nociceptive stimuli. We asked whether the mVPL can encode the intensity of sustained stimuli, and whether such coding arises from the anterolateral tract (ALT) or the dorsal column system. We recorded from the mVPL in urethane-anesthetized mice and lesioned the DCN to measure how mechanically-evoked responses change. The dorsal column system crosses just rostral to the DCN, whereas the ALT crosses within the spinal cord and runs laterally. Other spinothalamic and ascending neurons also cross within the spinal cord and run along the ventral aspect of the brainstem, and should be spared by lesioning the DCN. However, a potential caveat is that lesioning the DCN may disrupt the descending corticospinal tract, which passes just ventrally to the DCN toward the spinal cord. **a**, Schematized experimental configuration. A 32-channel multi-electrode array (MEA) was inserted into the middle VPL of the right thalamus. Mechanical indentation of the contralateral (left) hindpaw was performed before and after lesion of the left DCN by aspiration. The lesion should disrupt DCN input from reaching the VPL, but spare input from ALT neurons. **b**, Coronal section of the brain (50 μm) following recording with DiI labeling from the MEA in the right VPL (top). Scale bar is 1 mm. Inset shows magnified view of the thalamus (bottom left) and inferred location of VPL and thalamic reticular nucleus (Rt, bottom right). Scale bar is 1 mm (top) and 100 μm (bottom). Histological verification of the electrode track was performed in one animal. **c**, Coronal section of the brainstem showing the location of the lesion. The dorsal brainstem was aspirated at the level of the caudal DCN on the left side (top). Inset shows magnified view of the lesion site, and location of the gracile nucleus (GrN) that relays information from the hindpaw, and also the cuneate tract (cu) and some of the cuneate nucleus (CuN). Scale bar is 1 mm (top) and

200 μm (bottom). Histological verification of the lesion was performed in two animals. **d**, Waveform for a unit recorded across four channels of the MEA prior to the DCN lesion with a subset of single events (gray) and overlaid average (black). **e**, Autocorrelogram for unit shown in **d–o** prior to the lesion. **f**, Raster of firing for unit shown in **d–o** in response to a 300 ms step indentation of varying force. Trials were interleaved, but are shown sorted for presentation. **g**, Maximum firing rate for the onset of step indentation (0–50 ms) as a function of indentation force for unit shown in **d–o** prior to lesion. Dashed green line indicates the measured threshold. **h**, Average firing rate during the sustained component of the step indentation (150–300 ms) for unit shown in **d–o** prior to lesion as a function of force. **i**, Time course of firing rate for unit shown in **d–o** in response to indentation for 10 mN (top) and 250 mN (bottom) prior to lesion. **j–o**, Same as **d–i**, but for the same unit after DCN lesion. **p**, Average histograms for all mVPL units in response to 10 mN 300 ms indentation prior to lesion (left) and for the same units after DCN lesion (right). Units are sorted by strength of on-response to 10 mN indentation. **q**, Mean ± s.e.m of all mVPL units in response to 10 mN indentation before (left) and after (right) lesion of the DCN. **r**, Average histograms for all mVPL units in response to 250 mN 300 ms indentation prior to lesion (left) and for the same units after DCN lesion (right). Units are same as shown in **p**, but sorted by strength of sustained response at 250 mN. **s**, Mean ± s.e.m. of all mVPL units in response to 250 mN indentation before (left) and after (right) lesion of the DCN. **t**, Cumulative histogram of on-response threshold for the same units before (black) and after (red) lesion of the DCN. About 60% of units could no longer be activated by indentations up to 300 mN following DCN lesion. Kolmogorov-Smirnov test, p < 0.001. **u**, Mean ± s.e.m. of sustained response of all units as a function of indentation force before (black) and after (red) lesion of the DCN. Kolmogorov-Smirnov test, p < 0.001. **v**, Cumulative histogram of spontaneous firing rates for same units before (black) and after (red) lesion of the DCN. Kolmogorov-Smirnov test, p = 0.001. Data shown as n units in N animals (n/N).

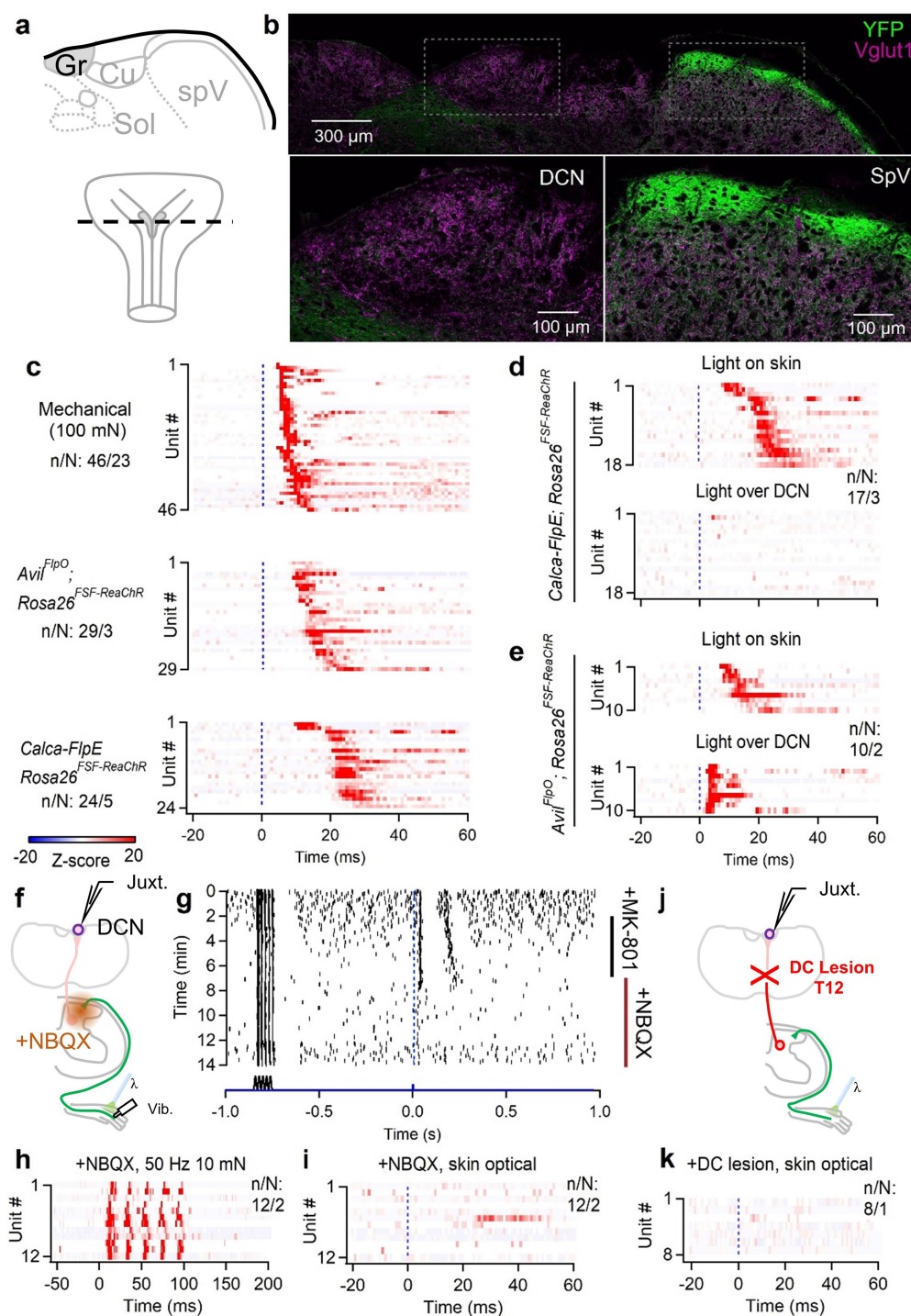

**Extended Data Fig. 8** | See next page for caption.

**Extended Data Fig. 8 | *Calca*+ HTMRs do not project directly to the DCN but activation of their endings in the skin drives long latency responses.** *Calca* is expressed in a broad set of sensory neurons. These include mechanically sensitive, polymodal receptors, and thermoreceptors that have A or C-fiber conduction velocity; *Calca*+ HTMRs have higher mechanical thresholds than Aβ-LTMRs[33–36]. *Calca*+ DRG neurons typically do not project to the DCN, although previous studies have shown ~10% of DRG neurons retrogradely labeled from the DCN express *Calca*[32]. Others have observed sparse *Calca*+ fibers within the DCN[37,38]. **a**, Schematic representation of a coronal section of brainstem containing the gracile and cuneate nucleus of the DCN (Gr, Cu), and the neighboring solitary nuclei (Sol) and spinal trigeminal nucleus (spV). The rostro-caudal area of the brainstem is shown at bottom. **b**, Immunofluorescence of Vglut1 (magenta) and ReaChR-YFP (green) in a brainstem section of a *Calca-FlpE; Rosa26*[FSF-ReaChR] animal showing the dorsal brainstem section corresponding to **a** (*top*). Insets of the gracile (*bottom left*) and spinal trigeminal (*bottom right*) are shown for comparison. YFP+ Fibers are densely labeled in the dorsal spinal trigeminal, but very few were detected in the gracile. Histology shown in **a-b** was performed in one animal. **c**, DCN neurons (random) were juxtacellularly recorded in the gracile and the latencies of responses to various stimuli were measured. DCN units had short latency responses to strong mechanical indentation beginning at time 0, in wildtype animals (*top*). In *Avil*[FlpO]; *Rosa26*[FSF-ReaChR] randomly recorded DCN units also had short latency responses to optical activation of sensory neurons in the skin (*middle*). In *Calca-FlpE; Rosa26*[FSF-ReaChR] animals optical activation of *Calca*+ endings in the skin evoked longer latency responses in randomly recorded DCN units (*bottom*). **d**, Random DCN units and their average response to light activation of *Calca*+ endings in the skin in *Calca-FlpE; Rosa26*[FSF-ReaChR] animals (*top*). In the same units, light was also delivered directly over the DCN (*bottom*). Light over the DCN evoked weak responses in 2/19 units. **e**, Random DCN units for *Avil*[FlpO]; *Rosa26*[FSF-ReaChR] animals in which all sensory neurons express ReaChR. Robust short-latency firing could be evoked in all recorded DCN units when light was delivered over the DCN (*bottom*). **f**, Random DCN neurons were recorded and light was delivered to the skin in *Calca-FlpE; Rosa26*[FSF-ReaChR] animals. The same spot on the skin was also stimulated using a gentle (10–20 mN) vibration (50 Hz, 100 ms). Synaptic blockers were then applied to the lumbar spinal cord to measure their effect on optically- and mechanically-evoked responses. **g**, Raster of a random DCN unit and its response to 50 Hz vibration and optical activation of *Calca*+ endings in the skin. After a baseline period, the NMDAR antagonist MK-801 was applied to the lumbar spinal cord (10 mM, 10 µL). After 5 min, the MK-801 was washed away with saline, and NBQX was added to the spinal cord (10 mM, 10 µL). **h**, Average histograms of vibration responses of random DCN units in which synaptic blockers were applied to the spinal cord. **i**, Same units shown in **h** and their response to optical activation of *Calca*+ endings in the skin following application of synaptic blockers to the spinal cord. **j**, Random units in the DCN were recorded in a *Calca-FlpE; Rosa26*[FSF-ReaChR] animal. The area of the gracile responding to the hindlimb was located and the DC at T12 was lesioned. Responses were then measured to optical activation of *Calca*+ endings in the skin. **k**, Average responses in random DCN units in hindlimb region of the gracile to optical activation of *Calca*+ endings in the skin after lesioning the DC at T12. The entire hindpaw was sampled for each unit to ensure the receptive field was not missed. Data shown as n units in N animals (n/N).

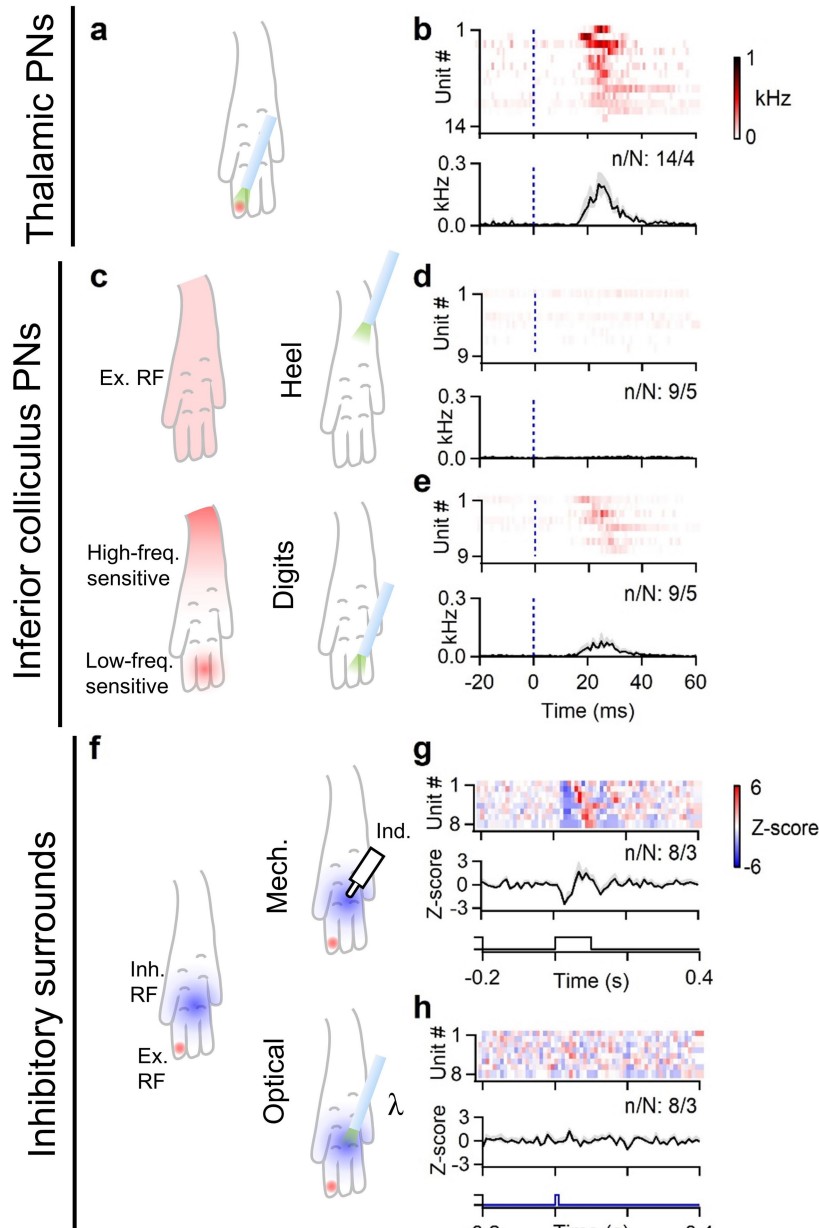

**Extended Data Fig. 9 | *Calca*⁺ HTMRs can drive IC-PNs, but not inhibitory surrounds.** All experiments were performed in *Calca-FlpE*; *Rosa26^FSF-ReaChR* animals. **a**, Schematic representation of a typical VPL-PN excitatory receptive field and location of light stimulus delivered. **b**, Average firing rate of identified VPL-PNs in response to optical activation of *Calca*⁺ endings in the skin (*top*). Average evoked firing rate across all VPL neurons following optical stimulation of *Calca*⁺ cutaneous endings (*bottom*). Bins are 1 ms. Color scale for firing rate shown at right. Histogram shown as mean ± s.e.m. **c**, The receptive fields of vibration-tuned IC-PNs typically span the entire paw (*top left*). However, with very low stimulus intensities, it was revealed that the heel is more sensitive to high-frequency vibration, whereas the toes are more sensitive to low frequency vibration (*bottom left*; see Extended Data Fig. 2). Light was thus delivered to either the heel (*top right*), or the most sensitive area of the digits (*bottom right*). **d**, Response of individual vibration-tuned IC-PNs in response to optical stimulation of *Calca*⁺ afferents in the heel (*top*), and average response across

units (*bottom*). Bins are 1 ms. Color scale for firing rate same as in **b**. Histogram shown as mean ± s.e.m. **e**, Response of individual vibration-tuned IC-PNs in response to optical stimulation of the digits (*top*), and average response across units (*bottom*). Units are the same as those shown in **d**. Histogram shown as mean ± s.e.m. **f**, The inhibitory surround of a DCN unit was stimulated mechanically using 10–20 mN indentation (*top right*). The same part of the inhibitory surround was then also stimulated optically (*bottom right*). **g**, Average responses of individual DCN units to 10–20 mN indentation delivered to their inhibitory surrounds (*top*), and average inhibitory response across all units (*bottom*). Bins are 10 ms. Color scale Z-score shown at right. Histogram shown as mean ± s.e.m. **h**, Average responses of individual DCN units to optical pulse delivered to their inhibitory surrounds (*top*), and average response across all units (*bottom*). Units are the same as those shown in **g**. Histogram shown as mean ± s.e.m. Data shown as n units in N animals (n/N).

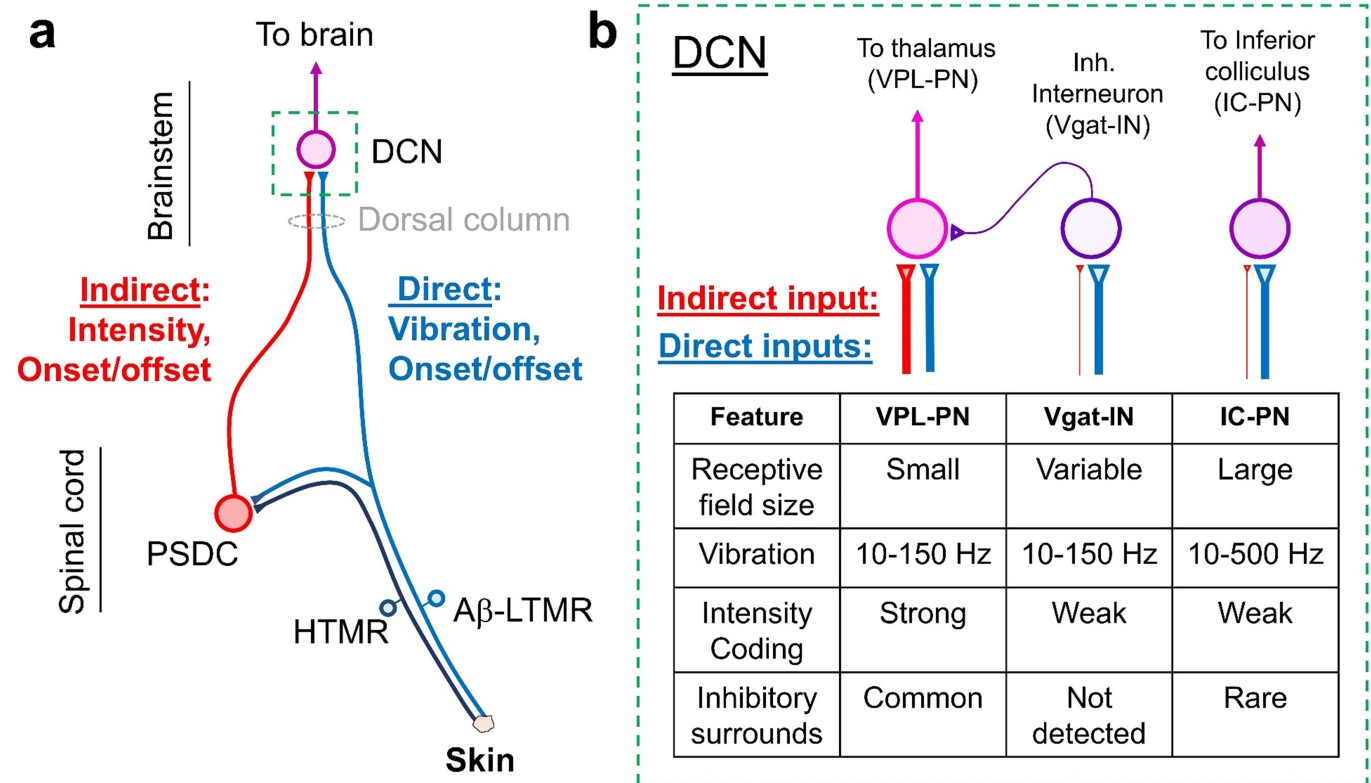

**Extended Data Fig. 10 | A model for sensory representation in the hindlimb dorsal column system. a**, Mechanical stimuli in a small area of skin are detected by LTMRs and HTMRs. LTMRs ascend the dorsal column and synapse onto neurons in the DCN, forming the direct pathway. LTMRs also form collaterals within the spinal cord dorsal horn. PSDCs in the spinal cord dorsal horn receive mechanical information from LTMRs and HTMRs, either directly or through local interneurons. PSDC axons ascend the dorsal column, forming the indirect pathway. PSDCs of the indirect pathway signal the intensity of sustained stimuli and the onset/offset of stimuli, as well as low-frequency vibratory stimuli. LTMRs of the direct pathway signal rapid movements and high-frequency vibratory stimuli, in addition to the onset/offset of stimulation. These two pathways converge somatotopically such that individual DCN neurons can encode multiple features in a small patch of skin. **b**, Different contributions from PSDCs and LTMRs drive distinct response properties in DCN neuron types. *Left*: VPL-PNs have small RF sizes and commonly have inhibitory surrounds driven by DCN Vgat-INs. Direct LTMR input drives phase-locked spiking up to ~150 Hz.

VPL-PNs can also commonly encode sustained stimuli at high forces mediated by indirect PSDC input. Thus, VPL-PNs receive prominent direct and indirect pathway input. *Middle:* Local Vgat-INs have varying receptive field sizes and can entrain to vibrations up to 150 Hz. They can, in some cases, be activated by high-frequency stimulation, but typically do not entrain their firing. Vgat-INs rarely fire in response to the sustained component of stimuli and thus poorly encode the intensity of sustained stimuli. Inhibitory surrounds in other DCN neurons that are generated by Vgat-INs are not strongly driven by indirect input, and are not driven by *Calca*+ HTMR activation. *Right:* The most common type of IC-PNs (vibration-tuned) have very large receptive field sizes and can entrain to high-frequency vibratory stimuli up to 500 Hz. These high-frequency responses are likely generated exclusively by direct LTMR (Aβ-RA2-LTMR) input. They can encode sustained stimuli, but not as well as VPL-PNs. They receive some input from HTMRs, likely through PSDCs, but only to a subset of their receptive field.

# Reporting Summary

## Statistics

For all statistical analyses, confirm that the following items are present in the figure legend, table legend, main text, or Methods section.

| n/a | Confirmed | |
|---|---|---|
| ☐ | ☒ | The exact sample size (*n*) for each experimental group/condition, given as a discrete number and unit of measurement |
| ☐ | ☒ | A statement on whether measurements were taken from distinct samples or whether the same sample was measured repeatedly |
| ☐ | ☒ | The statistical test(s) used AND whether they are one- or two-sided *Only common tests should be described solely by name; describe more complex techniques in the Methods section.* |
| ☒ | ☐ | A description of all covariates tested |
| ☐ | ☒ | A description of any assumptions or corrections, such as tests of normality and adjustment for multiple comparisons |
| ☐ | ☒ | A full description of the statistical parameters including central tendency (e.g. means) or other basic estimates (e.g. regression coefficient) AND variation (e.g. standard deviation) or associated estimates of uncertainty (e.g. confidence intervals) |
| ☐ | ☒ | For null hypothesis testing, the test statistic (e.g. $F$, $t$, $r$) with confidence intervals, effect sizes, degrees of freedom and $P$ value noted *Give P values as exact values whenever suitable.* |
| ☒ | ☐ | For Bayesian analysis, information on the choice of priors and Markov chain Monte Carlo settings |
| ☒ | ☐ | For hierarchical and complex designs, identification of the appropriate level for tests and full reporting of outcomes |
| ☒ | ☐ | Estimates of effect sizes (e.g. Cohen's *d*, Pearson's *r*), indicating how they were calculated |

*Our web collection on statistics for biologists contains articles on many of the points above.*

## Software and code

Policy information about availability of computer code

| Data collection | Electrophysiology data was collected using pClamp 11 (juxtacellular recordings) or Intan RHX 3.0.4 (Multi-electrode array recordings). Anatomical data was collected using Zen Blue or Zen Black (2012). Images of the paw and behavior were collected using Spinview 2.3.0.77. |
|---|---|
| Data analysis | Data were analyzed using custom scripts written in Matlab R2019a. Custom scripts used in this study are posted on Github (see Methods for link). Electrophysiology data collected using multielectrode arrays were analyzed using Kilosort 2.0 and Phy (Pachitaru et al. 2016; Rossant et al. 2016). Figures were generated in Igor 6.37. |

For manuscripts utilizing custom algorithms or software that are central to the research but not yet described in published literature, software must be made available to editors and reviewers. We strongly encourage code deposition in a community repository (e.g. GitHub). See the Nature Portfolio guidelines for submitting code & software for further information.

## Data

Policy information about availability of data

All manuscripts must include a data availability statement. This statement should provide the following information, where applicable:
- Accession codes, unique identifiers, or web links for publicly available datasets
- A description of any restrictions on data availability
- For clinical datasets or third party data, please ensure that the statement adheres to our policy

Source data are available with the published paper. Datasets generated in this study are available from the corresponding author upon reasonable request.

# Field-specific reporting

Please select the one below that is the best fit for your research. If you are not sure, read the appropriate sections before making your selection.

☒ Life sciences          ☐ Behavioural & social sciences          ☐ Ecological, evolutionary & environmental sciences

For a reference copy of the document with all sections, see nature.com/documents/nr-reporting-summary-flat.pdf

# Life sciences study design

All studies must disclose on these points even when the disclosure is negative.

| | |
|---|---|
| Sample size | Sample size was not predetermined. Sample sizes were based on previous studies from our lab and others that are common in the field (Choi et al. 2020; Lehnert et al. 2021; Petty et al. 2021). |
| Data exclusions | No data were excluded from analysis unless it failed to meet quality standards, except the following: For experiments in Extended Data Figure 7, units that had initially high thresholds at baseline and became mechanically insensitive following DCN lesion were not included for analysis because stimuli may have been off center of the receptive field. |
| Replication | We performed experiments with multiple animals to confirm reproducibility. All attempts at replication were successful. The number of replications is noted in Extended Data Table 1. |
| Randomization | Most experiments in this study did not compare separate groups, so randomization was not used. For pharmacology experiments, mice were randomized to either receive drug application or not. |
| Blinding | It was not possible to perform blind experiments in this study as mice and treatments were easily identifiable as the experiments were performed. |

# Reporting for specific materials, systems and methods

We require information from authors about some types of materials, experimental systems and methods used in many studies. Here, indicate whether each material, system or method listed is relevant to your study. If you are not sure if a list item applies to your research, read the appropriate section before selecting a response.

## Materials & experimental systems

| n/a | Involved in the study |
|---|---|
| ☐ | ☒ Antibodies |
| ☒ | ☐ Eukaryotic cell lines |
| ☒ | ☐ Palaeontology and archaeology |
| ☐ | ☒ Animals and other organisms |
| ☒ | ☐ Human research participants |
| ☒ | ☐ Clinical data |
| ☒ | ☐ Dual use research of concern |

## Methods

| n/a | Involved in the study |
|---|---|
| ☒ | ☐ ChIP-seq |
| ☒ | ☐ Flow cytometry |
| ☒ | ☐ MRI-based neuroimaging |

# Antibodies

| | |
|---|---|
| Antibodies used | PRIMARIES: mouse anti-NeuN, 1:1000 MAB377, Millipore, Clone A60 guinea pig anti-Vglut1, 1:2000 135301, Synaptic Systems SECONDARIES: goat anti-mouse Alexa-488, 1:500, ab150113, Abcam IB4-Alexa 647, 1:300, I32450, ThermoFisher FITC goat anti-GFP, 1:500, ab6662, Abcam |
| Validation | mouse anti-NeuN (1:1000, MAB377, Millipore): Abraira et al. Cell, 2017 guinea pig anti-Vglut1, (1:2000 135301, Synaptic Systems): Validated by vendor; performed western blot on brain homogenate of wildtype and vglut1 KO mice. |

# Animals and other organisms

Policy information about studies involving animals; ARRIVE guidelines recommended for reporting animal research

| | |
|---|---|
| Laboratory animals | This study uses mice. Mice were older than postnatal day 40 and were of either sex. The following mouse lines were used: C57Bl/J6; Calca-FlpE (Choi et al. Nature 2020); AvilFlpO (Choi et al. Nature 2020); AvilCre (Zhou et al. PNAS 2010); Cdx2-Cre (Coutaud & Pilon Genesis 2013); RosaLSL-Acr1 (Li et al. Elife 2019); RosaFSF-ReaChR (derived from Hooks |

et al. Journal of Neuroscience 2015); Animals were maintained on mixed C57Bl/J6 and 129S1/SvlmJ backgrounds. C57Bl/J6 were obtained from Jackson Laboratories.

Wild animals

This study did not involve wild animals.

Field-collected samples

This study did not involve field-collected samples.

Ethics oversight

All experimental procedures were approved by the Harvard Medical School Institutional Care and Use Committee and were performed in compliance with the Guide for Animal Care and Use of Laboratory Animals.

Note that full information on the approval of the study protocol must also be provided in the manuscript.

