## [Peer Review File · Nature]

Manuscript Title: The encoding of touch by somatotopically aligned dorsal column subdivisions

Reviewer Comments & Author Rebuttals

Reviewer Reports on the Initial Version:

Referees' comments:

Referee #1 (Remarks to the Author):

The gracile nucleus is a brainstem structure known to be important for touch sensation that receives input from large diameter mechanosensory neurons (A-LTMRs) and a subset of principle output neurons from spinal cord dorsal horn (PSDCs). The function and organization of this curious arrangement of direct and indirect touch inputs converging within this brainstem area is the subject of this fascinating new study by Turecek and colleagues. The study uses an elegant mix of classic approaches (in vivo juxacellular electrical recordings and pharmacology) and modern genetic strategies (optical inhibition/excitation of different classes of touch neurons). The data supports a model whereby the two pathways converge and convey different types of information: the direct pathway for acuity and the indirect pathway for intensity. The hypothesis is intriguing and supported by the provided evidence.

Overall, this manuscript was a joy to read. It is an important study and one that generates many new questions, some of which I provide below. While this work does not need substantial revisions or new experiments, there are areas worth considering/discussing further before publication, particularly if the authors have the information.

Comments/suggestions:

(1) I understand these are technically demanding experiments that would be near impossible to perform in awake mice. I also feel that the DCN recordings where the direct pathway is optically stimulated are likely very informative since these are likely monosynaptic connections. However, the anesthesia almost certainly is impacting the indirect pathway. The authors need to better acknowledge the limits of performing these experiments under anesthesia throughout the manuscript. There is substantial evidence that neuronal firing and stimulus encoding within the ascending somatosensory pathway is altered by anesthetics. In the awake state, most types of high threshold stimuli would also activate nociceptive pathways and generating profound behavioral reactions. I suspect the state of the information processing in the dorsal horn under these conditions would alter the information carried by the PSDCs, not to mention other polysynaptic routes of modulation in DCN. I feel it remains quite possible the intensity coding model is an oversimplification.

(2) The authors state they used pharmacological block of the spinal cord, as they have no genetic means of targeting the PSDC neurons / indirect pathway for loss of function experiments. The authors could however have targeted this pathway using a different viral approach. Injection of an

AAV encoding Cre into the lumbar spinal cord of LSL-Acr1 mice will lead to Acr1 expression in spinal cord neurons and axon terminals. Use of DRG-sparing serotypes and promoters (Haenraets, 2017), or through a Flp-off Cre virus in the Avil-flpO mouse, would avoid expression in sensory neurons by retrograde infection of central terminals. Light inhibition of the spinally originating terminals from PSDC neurons in the DCN would enable the authors to selectively silence the indirect pathway. Could the authors comment on why they did not consider such an intersectional viral-genetic strategy for accessing the indirect pathway? Such an approach could also be used for gain-of-function experiments using excitatory opsins. For example, presumably DCN neurons should respond in the high intensity range and in a sustained manner to PSDC terminal activation, but not to high frequency stimulation.

(3) Can more information about the connectivity of direct vs indirect pathways in the DCN be provided? Does the indirect pathway target inhibitory cells or only VLP projecting neurons? For example, some understanding of the projections would clarify if the indirect pathway contributes to the inhibitory surround. More importantly, this type of information might shed some light on how the two projections become topographically aligned. Slice recording would be ideal but almost certainly beyond the scope here, but some histology could add a lot.

(4) The IC projecting cells are only mentioned at the onset in passing and seem to be specialized for high frequency detection with low spatial resolution. Is there a function for the indirect pathway in these cells as well or is the indirect pathway specific for VPL projecting DCN neurons?

(5) The surround inhibition model relies on engaging the inhibitory neurons. Does pharmacological block of Gaba signaling then impair spatial acuity?

(6) Perhaps a detail, but in a previous paper this group generated a Bmp1r-Cre to selectively target Calca+ HTMRs. Curious why then here a Calca-Flp is used which broadly targets many types of thermal and polymodal nociceptors? The more broad optical activation of several classes of nociceptors in the Calca-Flp experiments should be made clear.

(7) Presumably NBQX blocks the Calca evoked responses in the DCN?

(8) It would be helpful if the number of cells were reported more clearly in each figure and the legends written with a bit more detail. Understandably, many experiments have recordings from only a handful of cells. However, sometimes, it took effort to distinguish data from an example neuron as compared to the sum of cells recorded.

(9) Proprioceptor axons also project to the gracile via the dorsal column (e.g. PMID: 24198362) and these neurons respond to vibration stimuli. To what extent might these neurons be contributing to the vibration responses and should their contribution be acknowledged?

Referee #2 (Remarks to the Author):

Comments to Author:

Here Turecek et al., use in vivo electrophysiological recordings together with genetic mouse lines to study functional synaptic connectivity relationships between direct and indirect pathways innervating the dorsal column nuclei (DCN). It was already known that DCN encode discriminatory touch, vibration, and intensity, but the relative contributions of the direct and particularly the PSDC inputs to touch sensation were not entirely clear.

The authors report that direct LTMR inputs to DCN convey vibrotactile stimuli with high temporal precision, while indirect inputs from PSDC neurons transmit touch onset and the intensity of sustained mechanical input in the high force range. The authors attribute the encoding of high intensity sustained mechanical stimuli by PSDC neurons to the indirect pathway uniquely receiving HTMR input. Under normal conditions, cutaneous responses carried by the two pathways re-align in the DCN and conserve somatotopy. The authors also describe the response properties of inhibitory interneurons of the DCN as well as two distinct excitatory neuron types that have unique postsynaptic targets (VPL or IC). These excitatory neuron types are identified by antidromic activation from the target areas.

A surprise was the similar number of direct and indirect inputs to DCN per spinal segment which is shown in a supplementary figure and calculated by one method. Based on the representation of Abeta LTMR in the DRG, would the authors expect more than ~120-150 neurons (out of ~3200 myelinated neurons per DRG) to be back-labeled from the DCN? Did I interpret this calculation incorrectly?

Major comments

Although the manuscript does provide new information, novel contributions to our understanding of DCN coding and particularly the role of the PSDCs provided here is too limited (does not reach a level of significance expected) for publication in Nature. The findings on the contributions of DRG inputs are as expected based on decades of work and the findings on the contributions of PSDC are convincing yet preliminary in scope. The lack of genetic tools to selectively manipulate the indirect pathway (other than the Calca mice) meant the findings relied on a confirmation through lesions or silencing with NBQX in the spinal cord, which are imprecise methods. Lastly, analysis of this circuitry should include the molecular identification of the DCN and PSDC cell types and a histological examination of the direct and indirect inputs to these DCN cells.

Other comments:

1. Acknowledging that there are character/word limits, the figure legends nevertheless need to provide more detail /better inform the reader of experimental paradigms performed in each panel.
2. What is the rationale for using Kolmogorov-smirnov instead of shapiro-wilk test when the number of units is less than 50 in most experiments? The results are unlikely to change significantly, but the reason behind the choice should be clarified.
3. Throughout the manuscript, authors describe intensity of stimuli as “strong”, “intense”. It would benefit the readers if the authors can clarify early in the manuscript the range of pressure/intensity that corresponds to such descriptive words. Is the high intensity stimulation noxious/painful? The

authors avoid tissue damage, but the stimuli might nevertheless be noxious (i.e. elicit a nocifensive behavior in awake mice). A behavioral control is needed.

4. In figures for receptive field mapping – there are different Hz for each DCN unit RF mapping (ex. Fig 4. 100, 60, 200, 500 Hz), however the legend description says that the vibration was set to one frequency. Please clarify.

5. Experiments in Figure 2 show that direct dorsal column pathway is a driver of inhibitory surround in PN that project to VPL, and that these neurons do not respond to HTMR input. Without the ability to selectively manipulate the PDSCs, a role for these cells in surround inhibition nevertheless cannot be ruled out.

6. Figure 2 should show how inhibition of direct pathway inputs to DCN affects the DCN response to high forces (100~300 mN).

7. For recording experiments of the DCN, authors do not indicate the location of their recording relative to the rostral-caudal axis of the DCN. These segments have different RF and response properties. Post hoc histology or images of the recording sites should be included. Furthermore, for the lesion studies, the lesion location seems to be caudal part of DCN and part of the cervical spinal cord/ TG system. Have the authors lesioned across the entire DCN to make sure their lesion effects are not site specific. Lastly, for antidromic activation confirmation, images of electrode path would be useful.

8. In the main text, the authors claim (line 153-154) that no CGRP expressing neurons contribute to the direct pathway to the DCN referring to a paper the lab previously published. A more precise statement should be made or additional references that support the claim should be included since the noted reference suggests there are sparse CGRP+ projections from DRG to DCN.

9. To support the statement of line 166-167, graphs that show the percentage of units responding and not responding to mechanical stimuli should be shown.

10. The title is ambiguous and could just as well suggest that the authors will report that different areas of the DCN encode various features of touch. Similarly, the last sentence of the summary expresses concepts in a way that is so vague it makes them not surprising and further the descriptions of functional/biological purpose are also so vague that they essentially lack meaning and rather give the impression that not much was learned in the end from performing these analyses.

11. Line 46-49, would be good to make clear that the three DCN neuron types studied here comprise a majority of the neurons in the DCN but that other DCN PN populations exist (i.e spinal cord projecting neurons, PAG projecting neurons etc).

12. It would be helpful if there is a graphic representation that shows the quantified latencies of responses (line 159-160).

13. The phrasing used in the text: that activation of PN-VPL by optical activation in the skin did not show activation when optically activated in the DCN begs the question of whether optical activation

of Calca+ afferents in DCN causes firing of any DCN neurons. I understand the prediction that if the PN-VPLs activated by skin are directly connected to Calca+ afferents from DRG, then you would expect they would be activated in DCN. However, any of the responses of DCN neurons to optical activation of Calca+ afferents in DNC should be reported.

14. Why is the x-axis of Extended Data Fig. 7T shown as 100 mN max force but the figure legend discusses 300 mN?

15. Line 465, in Extended Data Fig. 2, the title may be changed to projection neuron subtypes, not just IC PN since there are experiments addressing other subtypes. Also need more clear statement of what the experiment is. Was it only recorded from IC PN and then generalize the types to VPL PN? Or recorded from random neurons in the DCN?

16. Line 639, in Extended Data Fig. 7, there is a run-on sentence

17. Line 702-703 mention that the corticospinal tract will be perturbed. Would be good to mention the same thing in Extended Data Fig. 7 and mention in the main text

Referee #3 (Remarks to the Author):

The coding of tactile information has been extensively studied in the peripheral mechanoreceptors and in the primary somatosensory cortex. Recently, it has become clear that processing and integration of tactile information occurs at the intermediate target brain regions as primary mechanosensory information ascends the somatosensory neuroaxis. Mechanosensory signals for discriminative touch perception reach the brain via the spinal cord dorsal column directly from peripheral mechanoreceptors that innervate the dorsal column nuclei (DCN) of the brainstem. A second indirect pathway formed by post-synaptic dorsal column neurons (PSDCs) within the spinal cord dorsal horn, which receive input from mechanoreceptors, also project to the DCN. The significance and relative contributions of these two major DCN inputs to the coding of touch is still unclear. The work now submitted, importantly sheds light on how signals encoding different tactile features are differentially conveyed by the two independent pathways to converge in the DCN. The authors propose that the LTMRs and the direct dorsal column pathway underlie vibration tuning, whereas PSDC neurons and the indirect dorsal column convey the intensity of sustained stimuli, which constitutes an important and interesting contribution to the field.

Comments:

- After an initial characterization of the different DCN neuron types, where the authors describe that the most abundant ventral posterolateral thalamus VPL projecting neurons (VPL-PNs) have small excitatory receptive fields with large regions of surround suppression, opposed to inferior colliculus projecting neurons (IC-PNs) that have larger and exclusive excitatory receptive fields, they addressed how the unique response properties of DCN neuron subtypes are generated. Experiments shown in Figures 2A-J lead to the conclusion that the indirect pathway contributes to responses elicited by low-frequency (10Hz) mechanical stimuli, while high-frequency vibration (50 to 300Hz) is exclusively encoded by the direct dorsal column pathway. Since VPL-PNs seem to have a high degree of spatial information encoding due to their inhibitory surround receptive fields, the authors asked how the direct and indirect pathways contribute to this. Using a 50Hz vibration stimulation protocol, while selectively inhibiting the direct pathway, resulted in the suppression of the inhibitory surround of DCN neurons leading to the conclusion that the direct dorsal column pathway is the primary driver of inhibitory surround receptive fields in VPL-PNs. In order to test if the indirect pathway would also have a role in driving the inhibitory surround of VPL-PNs receptive fields, this reviewer suggests to perform the experiment 2K-L also with low frequency stimulation, since PSDC neurons do not seem to elicit a DCN response at high frequency stimulations. Also, as shown in Figure 3, PSDC neurons seem to mediate force intensity tuning in the DCN. I wonder if indentation stimulation is also able to elicit surround inhibition in DCN neurons. If so, I think it would be relevant to test this paradigm and explore the potential role of the indirect pathway to the spatial tuning of VPL-PNs.

- Figure 2H shows that silencing LTMR axon terminals to block direct pathway inputs, was not able to eliminate DCN responses upon 10Hz mechanical stimuli, leading to the conclusion that the indirect pathway was able to convey signals from low-frequency mechanical stimuli to generate responses in the DCN. This evidence is based on a relative low sample number with a p value of 0.056. Since this is a relevant information, I think this should be better corroborated with a more robust data set.

- Figures 3D-M explore the contribution of indirect pathway to DCN representation of stimulus intensity. The authors nicely show that when excitatory transmission was blocked in the spinal cord, suppressing PSDC input, firing during the sustained phase of indentation was eliminated specially at high force ranges, pointing to an interesting observation that in the absence of PSDC input, DCN neurons are no longer able to encode the intensity of the maintained stimuli. To corroborate this observation I think this experiment should also be done when silencing the direct pathway (Avil::Acr1 experiment), as in this experiment only low forces stimuli were applied (5-10mN) and analysis such as shown in panels F-M should be included.

Author Rebuttals to Initial Comments:

Referees' comments:

Referee #1 (Remarks to the Author):

The gracile nucleus is a brainstem structure known to be important for touch sensation that receives input from large diameter mechanosensory neurons (A-LTMRs) and a subset of principle output neurons from spinal cord dorsal horn (PSDCs). The function and organization of this curious arrangement of direct and indirect touch inputs converging within this brainstem area is the subject of this fascinating new study by Turecek and colleagues. The study uses an elegant mix of classic approaches (in vivo juxacellular electrical recordings and pharmacology) and modern genetic strategies (optical inhibition/excitation of different classes of touch neurons). The data supports a model whereby the two pathways converge and convey different types of information: the direct pathway for acuity and the indirect pathway for intensity. The hypothesis is intriguing and supported by the provided evidence.

Overall, this manuscript was a joy to read. It is an important study and one that generates many new questions, some of which I provide below. While this work does not need substantial revisions or new experiments, there are a areas worth considering/discussing further before publication, particularly if the authors have the information.

Comments/suggestions:

(1) I understand these are technically demanding experiments that would be near impossible to perform in awake mice. I also feel that the DCN recordings where the direct pathway is optically stimulated are likely very informative since these are likely monosynaptic connections. However, the anesthesia almost certainly is impacting the indirect pathway. The authors need to better acknowledge the limits of performing these experiments under anesthesia throughout the manuscript. There is substantial evidence that neuronal firing and stimulus encoding within the ascending somatosensory pathway is altered by anesthetics. In the awake state, most types of high threshold stimuli would also activate nociceptive pathways and generating profound behavioral reactions. I suspect the state of the information processing in the dorsal horn under these conditions would alter the information carried by the PSDCs, not to mention other polysynaptic routes of modulation in DCN. I feel it remains quite possible the intensity coding model is an oversimplification.

We agree that this is an important consideration.

We performed new experiments in which we observed the response of awake head-fixed mice to our most intense stimulus delivered in this study (300 mN using a 1 mm blunt probe). Animals rarely withdrew their paw or exhibited signs of pain, such as kicking or licking. In contrast, brief activation of CGRP⁺ neurons using light evoked rapid paw withdraw and kicking. Thus, we think that the most intense mechanical stimuli used in this study are strong, but not noxious. Our results suggest that PSDCs may encode the intensity of stimuli that are not necessarily painful. These new results are shown in Extended Data Figure 6 and Supplemental Video 1.

Work performed by others would suggest that the responses of DCN neurons in awake animals can be modulated to some extent, but many of the features seen in unanesthetized animals are also seen under anesthesia. The distribution of receptive field sizes, the ability of DCN neurons to entrain to very

high-frequency vibration, sustained responses, and surround inhibition are all present in unanesthetized preparations and appear qualitatively similar (Bengtsson... Jorntell, 2013; Douglas, Ferrington & Rowe, 1978; Ferrington, Horniblow & Rowe, 1987). We have made the assumption that because DCN response properties are similar, PSDCs and LTMRs would also have similar contributions between awake and anesthetized animals.

The responses to innocuous stimuli in the spinal dorsal horn may also be similar in awake animals, and may even be more suppressed when the animal is awake due to descending inhibition (e.g. Wall, Freeman and Major, 1967; Collins, 1984). For innocuous stimuli, the effects of anesthesia seem to be much more profound in thalamus and cortex, where anesthesia can strongly suppress evoked responses, and the baseline activity is also very sensitive to the depth of anesthesia.

That said, we have not addressed this point directly. Our new experiments and the relevant literature do not rule out the possibility that PSDCs have other or additional response properties when the animal is awake. We have therefore acknowledged this possibility on page 10, and discuss it in greater depth in Extended Data Figure 1.

(2) The authors state they used pharmacological block of the spinal cord, as they have no genetic means of targeting the PSDC neurons / indirect pathway for loss of function experiments. The authors could however have targeted this pathway using a different viral approach. Injection of an AAV encoding Cre into the lumbar spinal cord of LSL-Acr1 mice will lead to Acr1 expression in spinal cord neurons and axon terminals. Use of DRG-sparing serotypes and promoters (Haenraets, 2017), or through a Flp-off Cre virus in the Avil-flpO mouse, would avoid expression in sensory neurons by retrograde infection of central terminals. Light inhibition of the spinally originating terminals from PSDC neurons in the DCN would enable the authors to selectively silence the indirect pathway. Could the authors comment on why they did not consider such an intersectional viral-genetic strategy for accessing the indirect pathway? Such an approach could also be used for gain-of-function experiments using excitatory opsins. For example, presumably DCN neurons should respond in the high intensity range and in a sustained manner to PSDC terminal activation, but not to high frequency stimulation.

Thank you for this suggestion. We avoided viral tools for several reasons:

When beginning this study, we found that input to the DCN is extremely strong. Silencing only a fraction of inputs would not have a meaningful or interpretable effect in DCN recording experiments. This required 1) a tool that very efficiently silences when it is expressed, and 2) very broad expression of the tool. We have found that viral tools are extremely useful for activation experiments, as not many cells need to be labeled in order to see an effect. On the other hand, we've found viral approaches to be much more challenging for silencing experiments because the majority of input neurons need to be infected in order to see an observable difference post-synaptically. Injections into the spinal cord with virus are generally inefficient, in our hands. As PSDCs make up <1% of the dorsal horn neurons, we reasoned that it would be unlikely that we could silence a meaningful fraction. In addition, the areas of spinal cord receiving input from the hindlimb are also spread over three segments, and delivering enough virus over three segments poses a major technical challenge.

Retrograde injections into the DCN are relatively easy to perform, but the efficiency can also be very low with AAV. Injecting virus also causes an immune/gliial response that alters the DCN when

recording several weeks following injection. Units are less abundant and we were concerned about interpreting any findings we would encounter were we to take a viral approach.

Thus, we think many of the strategies suggested would be useful for activating subsets of sensory neurons or PSDCs, but in our hands viral approaches have major limitations for silencing experiments in the DCN.

We did perform gain of function experiments early on using excitatory opsins restricted to lumbar dorsal horn neurons, but we found that these experiments were not as informative as we had hoped. These experiments indicated whether a given DCN neuron received input from PSDCs, but it gave no information about the type of information that was being conveyed. We found that activation of terminals in the DCN is generally very different than mechanical stimulation, and the two are difficult to relate to each other. Therefore, we ultimately focused on the silencing methods presented in the paper because they indicated what stimulus features PSDCs and LTMRs deliver to the DCN.

(3) Can more information about the connectivity of direct vs indirect pathways in the DCN be provided? Does the indirect pathway target inhibitory cells or only VLP projecting neurons? For example, some understanding of the projections would clarify if the indirect pathway contributes to the inhibitory surround. More importantly, this type of information might shed some light on how the two projections become topographically aligned. Slice recording would be ideal but almost certainly beyond the scope here, but some histology could add a lot.

We appreciate this comment and agree that it represents an interesting direction. We have chosen to spend time on *in vivo* electrophysiology as much as possible in the present study because it provides the most information about responses specific to different types of mechanical stimuli as well as the strength of inputs in relation to postsynaptic spiking. We agree that histology and slice physiology in many cases is informative, however tools for selective and simultaneous labeling of combinations of DCN PNs and INs as well as PSDC and multiple types of LTMR inputs needed for these approaches are limited or not yet available. We are envisioning future work using new, more advanced genetic tools, and histology and slice electrophysiology to determine the circuit mechanisms that explain the *in vivo* findings reported here.

(4) The IC projecting cells are only mentioned at the onset in passing and seem to be specialized for high frequency detection with low spatial resolution. Is there a function for the indirect pathway in these cells as well or is the indirect pathway specific for VPL projecting DCN neurons?

We think that this is great point. We have therefore performed new experiments to address the contribution of PSDCs to IC-PN responses. Our new findings indicate that very low frequency vibratory stimuli can be conveyed by PSDCs across DCN cell types, but high-frequency vibration is exclusively conveyed by the direct pathway (Extended Data Fig. 4). Vibration-tuned IC-PNs also receive HTMR input similar to VPL projecting neurons, but only in a subset of their receptive fields, typically on the digits (Extended Data Fig. 9). Activation of *Calca*⁺ endings in the skin can evoke responses in IC-PNs, but this activation is weaker than what we have seen in VPL projection neurons, consistent with their weaker responses to high intensity stimuli. These new findings are discussed on page 8 of the revised manuscript.

(5) The surround inhibition model relies on engaging the inhibitory neurons. Does pharmacological block of Gaba signaling then impair spatial acuity?

This is an interesting question. Blockade of GABA-Rs in the DCN can expand the receptive field within the DCN (See Schwark... Fuchs 1999). The prediction would be that this manipulation would also impair spatial detail represented in downstream regions, however we think that this would be more appropriately addressed in future studies that also use slice electrophysiological approaches.

(6) Perhaps a detail, but in a previous paper this group generated a *Bmp1r-Cre* to selectively target *Calca+* HTMRs. Curious why then here a *Calca-Flp* is used which broadly targets many types of thermal and polymodal nociceptors? The more broad optical activation of several classes of nociceptors in the *Calca-Flp* experiments should be made clear.

We used a broad line because we do not yet know the genetic identity of glabrous skin-innervating HTMRs that mediate high-threshold responses in the DCN. The *Bmp1r-Cre* mouse that we previously reported labels a population of $A\delta$ -HTMRs that innervates trunk hairy skin. We are currently characterizing the response properties of glabrous skin-innervating sensory neurons labeled by *Bmp1r-Cre* and other new genetic tools. This analysis will be the focus of a future study that will include much more detail about the physiological, morphological, and synaptic properties of genetically labeled $A\delta$ -HTMR subtypes that innervate glabrous skin of the paws and their contributions to somatosensory behaviors.

We have adjusted the main text and Extended Data Figure 8 to emphasize that *Calca-Flp* labels HTMRs, as well as some polymodal C-fiber neurons and thermoreceptors.

(7) Presumably NBQX blocks the *Calca* evoked responses in the DCN?

This is an excellent question and a clear prediction of our model. We have performed additional experiments in order to determine whether *Calca*-evoked responses in the DCN can be blocked by inhibiting spinal transmission. We found that adding synaptic blockers to the lumbar spinal cord strongly suppressed *Calca*⁺ sensory neuron ending-evoked responses in the DCN. These new experiments are shown in Extended Data Figure 8 and mentioned on page 8 of the revised manuscript.

(8) It would be helpful if the number of cells were reported more clearly in each figure and the legends written with a bit more detail. Understandably, many experiments have recordings from only a handful of cells. However, sometimes, it took effort to distinguish data from an example neuron as compared to the sum of cells recorded.

We have moved the number of experiments/animals into the figures themselves and have added more detail in the figure legends.

(9) Proprioceptor axons also project to the gracile via the dorsal column (e.g. PMID: 24198362) and these neurons respond to vibration stimuli. To what extent might these neurons be contributing to the vibration responses and should their contribution be acknowledged?

It is thought that proprioceptor projections to the gracile are rare or absent. In fact, one reason we focused on the gracile is that it has been shown to mostly lack proprioceptor input. We think that the

paper referred to might be labeling glabrous Ret⁺ LTMRs that are PV⁺; in that paper, mice were administered tamoxifen at E16.5-E17.5, when there are still PV⁺ cutaneous neurons that are also Ret⁺ (see Fig. 4B). When performing sparse labeling so that proprioceptors could be distinguished based on their anatomical properties, the same paper found that no lumbar proprioceptors ascended to the gracile nucleus (see Fig. 13J). We think that this result in the Luo et al. paper and other electrophysiological evidence (See Whitsel, Petrucelli & Sapiro, 1969; Fern, Harrison & Riddell, 1987) make a strong case that few proprioceptors, if any, reach the gracile. Consistent with this conclusion, in our hands units in the gracile are not responsive to limb movement if their cutaneous receptive fields on the hindpaw are unaltered. This is in contrast to the cuneate where we have observed clear responses to limb movement and muscle stretch.

Referee #2 (Remarks to the Author):

Comments to Author:

Here Turecek et al., use in vivo electrophysiological recordings together with genetic mouse lines to study functional synaptic connectivity relationships between direct and indirect pathways innervating the dorsal column nuclei (DCN). It was already known that DCN encode discriminatory touch, vibration, and intensity, but the relative contributions of the direct and particularly the PSDC inputs to touch sensation were not entirely clear.

The authors report that direct LTMR inputs to DCN convey vibrotactile stimuli with high temporal precision, while indirect inputs from PSDC neurons transmit touch onset and the intensity of sustained mechanical input in the high force range. The authors attribute the encoding of high intensity sustained mechanical stimuli by PSDC neurons to the indirect pathway uniquely receiving HTMR input. Under normal conditions, cutaneous responses carried by the two pathways re-align in the DCN and conserve somatotopy. The authors also describe the response properties of inhibitory interneurons of the DCN as well as two distinct excitatory neuron types that have unique postsynaptic targets (VPL or IC). These excitatory neuron types are identified by antidromic activation from the target areas. A surprise was the similar number of direct and indirect inputs to DCN per spinal segment which is shown in a supplementary figure and calculated by one method. Based on the representation of Abeta LTMR in the DRG, would the authors expect more than ~120-150 neurons (out of ~3200 myelinated neurons per DRG) to be back-labeled from the DCN? Did I interpret this calculation incorrectly?

The number of labeled DRG neurons in our experiments is an underestimate of the number of neurons projecting to the DCN. Although there are many myelinated neurons in the DRG, many will not project to the gracile. This includes A δ -LTMRs, A δ -HTMRs, and proprioceptors (see the response to point 9 above). We know that A β RA-LTMRs, A β Field-LTMRs and most A β SA-LTMRs project to the gracile. These populations together make up ~10-15% of the DRG. If there are ~5000 neurons in a mouse thoracic DRG, we would expect ~500-750 neurons labeled per DRG, which is more than what we observed. This difference is because the injections we performed were relatively small in order to prevent off-target labeling within the spinal cord. The DCN in mice is very small, and large volume injections of CTB can leak or spill over into other fiber tracts, retrogradely labeling neurons near the central canal, superficial dorsal horn, or deeper areas of the cord. Our intent was to keep the injections small in order to prevent counting cells that were not actually PSDCs.

Our goal was to obtain relative proportions of dorsal column axon types. A concern may be that the tracers we used have a biased selectivity for different cell types, for example preferentially labeling

PSDCs over LTMRs. For this reason, we performed experiments using different tracers, CTB and retrobeads, in order to see if they would yield similar results. In unpublished experiments, we have also used pseudo-rabies and AAV injections into the gracile to label these populations. We have seen similar ratios of labeled cells with these different tools. The primary difference we have noticed across these tracers are the different general degrees of uptake across all cell types (cholera toxin and retrobeads being the most efficiently uptaken). We now mention in the revised paper in Extended Data Figure 3 that small injections and multiple tracers were used to avoid off target labeling and for discerning the relative, rather than absolute numbers of DCN-projecting neuron types.

Major comments

Although the manuscript does provide new information, novel contributions to our understanding of DCN coding and particularly the role of the PSDCs provided here is too limited (does not reach a level of significance expected) for publication in Nature. The findings on the contributions of DRG inputs are as expected based on decades of work and the findings on the contributions of PSDC are convincing yet preliminary in scope. The lack of genetic tools to selectively manipulate the indirect pathway (other than the *Calca* mice) meant the findings relied on a confirmation through lesions or silencing with NBQX in the spinal cord, which are imprecise methods. Lastly, analysis of this circuitry should include the molecular identification of the DCN and PSDC cell types and a histological examination of the direct and indirect inputs to these DCN cells.

We agree with the reviewer that molecular identification of DCN and PSDC cell types is important. Our group has many years of experience genetically identifying sensory neuron and spinal dorsal horn neuron subtypes, and developing tools to study them (Luo et al 2009, Li et al, 2011, Liu et al 2012, Rutlin et al., 2014, Bai et al., 2015, Abraria et al., 2017, Neubarth et al., 2020, Sharma et al., 2020, Choi et al., 2020). Molecular identification of PSDC neurons for the purpose of genetic access has been an ongoing effort in the lab for over a decade, and there has so far been little success. PSDCs are very fragile, and are few in number making them difficult to isolate for sequencing or resolve in large scale sequencing datasets. We are actively working to identify PSDC subtypes, but even if it were complete, this effort would be difficult to fit into the current study.

With or without a molecular characterization, we think that our findings are substantial and would benefit the broader scientific community by being published in Nature. There are only a small number of studies that have examined PSDCs, and none that we are aware of define their role in shaping physiological responses to tactile stimuli in downstream regions of the somatosensory system.

Our findings are also important because many of them were surprising to us, and we think they will be to others. They have changed how we think about the encoding of tactile information in the dorsal column pathway. We did not expect to find that high frequency (>300 Hz) vibration is strongly biased from the DCN to the IC rather than the VPL. Moreover, although one might predict high frequency (>300 Hz) vibration to be conveyed by the direct pathway, we were surprised to see that PSDCs do not provide vibratory information above 10 Hz, despite reports that dorsal horn neurons can be excited by higher frequencies (e.g. Salter & Henry, 1990). We also did not expect to find that most thalamic-projecting DCN neurons would respond to activation of *Calca*⁺ endings in the skin, that PSDCs convey high-intensity stimuli to the DCN, or that this is propagated from the DCN to the thalamus rather than through the anterolateral pathway. We also predicted that PSDCs would contribute to surround inhibition in the DCN. However, for the stimuli we applied, we have not found that PSDCs drive inhibitory surrounds as we had hypothesized at the onset of the study. We certainly did not predict that the direct and indirect dorsal column pathways would be somatotopically aligned

with the degree of precision observed. This collection of new observations support an entirely new view of the encoding of touch in the dorsal column pathway.

Although some of the manipulations are not specific to PSDCs, we think that in the context in which they are used these manipulations are sufficient for the questions we set out to answer and the conclusions reached. We used a combination of classic techniques such as lesions and pharmacology, but we also used optogenetic silencing and activation strategies that have very recently been developed, and that we pushed further. We think that the findings are important regardless of the methods used to establish them, as long as the experiments are convincing and the conclusions are properly supported by evidence.

Other comments:

1. Acknowledging that there are character/word limits, the figure legends nevertheless need to provide more detail /better inform the reader of experimental paradigms performed in each panel.

Thank you for this valuable suggestion. We have added more detail to make each of the figure legends and figures more clear.

2. What is the rationale for using Kolmogorov-smirnov instead of shapiro-wilk test when the number of units is less than 50 in most experiments? The results are unlikely to change significantly, but the reason behind the choice should be clarified.

The Kolmogorov-Smirnov test is a widely used nonparametric method for comparing two samples. It is integrated into many statistics toolboxes and it is relatively well-known and recognizable. We thought that this made it a useful statistical test for much of our data, and one that readers would quickly recognize and have an understanding of its strengths and limitations. The Shapiro-Wilk test is used for testing normality, and we are not aware of its use as a two-sample test. We are very open to performing the test, but we have been unable to find a two-sample version that is readily available. The differences we see in our data are robust to multiple nonparametric statistical tests, and we agree that the interpretation of the results is unlikely to change using a Shapiro-Wilk test. Therefore, we have not changed the statistical tests presented.

3. Throughout the manuscript, authors describe intensity of stimuli as “strong”, “intense”. It would benefit the readers if the authors can clarify early in the manuscript the range of pressure/intensity that corresponds to such descriptive words. Is the high intensity stimulation noxious/painful? The authors avoid tissue damage, but the stimuli might nevertheless be noxious (i.e. elicit a nocifensive behavior in awake mice). A behavioral control is needed.

We agree with the reviewer that this is an important point. The strongest indentation used (300 mN, 1 mm probe tip) is not at all painful to us when applied to our finger tips, but it was unclear whether it could cause pain in mice that have thinner skin. Therefore, we performed additional experiments using awake mice to determine whether our stimuli are noxious/painful. We delivered the highest intensity indentation (300 mN, 1 mm probe tip) to awake head-fixed mice, and compared their response to optical activation of CGRP⁺ endings. We found that mice rarely withdrew their hindlimbs in response to 300 mN indentation, and many times had no reaction. In contrast, optical activation of CGRP⁺ endings consistently resulted in withdraw and kicking. We have included these additional experiments in Extended Data Figure 6 and Supplemental Video 1. We have stated on page 6 of the revised manuscript that the range of mechanical stimuli we are delivering is strong but not noxious.

4. In figures for receptive field mapping – there are different Hz for each DCN unit RF mapping (ex. Fig 4. 100, 60, 200, 500 Hz), however the legend description says that the vibration was set to one frequency. Please clarify.

These numbers refer to the maximal firing rate of the presented unit – we have changed the labeling in the revision to make this less confusing.

5. Experiments in Figure 2 show that direct dorsal column pathway is a driver of inhibitory surround in PN that project to VPL, and that these neurons do not respond to HTMR input. Without the ability to selectively manipulate the PSDCs, a role for these cells in surround inhibition nevertheless cannot be ruled out.

We have done additional experiments to address the contribution of PSDCs to inhibitory surrounds. Our results show that in the absence of direct LTMR input, the stimuli used can no longer drive inhibitory surrounds (Fig. 2). This indicates that PSDCs are unable to drive inhibitory surrounds evoked by indentation, or brief 10 and 50 Hz vibration.

We agree that this does not rule out all potential contributions of PSDCs to other forms of surround inhibition that we may not have tested, or other forms of tonic inhibition or inhibition related to the state of the animal. However, for the stimuli we have tested, it appears that PSDCs cannot drive inhibitory surrounds when direct LTMR input is silenced. We have made adjustments to the text on page 10 of the revised manuscript to state that PSDCs may contribute to other forms of inhibition and that our conclusions only apply to the stimuli we tested.

6. Figure 2 should show how inhibition of direct pathway inputs to DCN affects the DCN response to high forces (100~300 mN).

We initially set out to perform these experiments, but we were unable to adequately silence ascending input to the DCN when stimulating the skin at high forces. We delivered 100 mN indentation to skin in *Cdx2-Cre; Rosa26^{LSL-Acr}* animals in which all input to the DCN should be suppressed with light. However, 100 mN indentation activates such a powerful physiological response that it was capable of evoking some postsynaptic responses in DCN neurons even in the presence of light. In pilot experiments, we found that Acr was able to silence about 90% of synaptic release. When low-threshold stimuli were delivered, we think the remaining amount is incapable of evoking responses in the DCN. However, when we deliver intense stimuli, the remaining 10% of a large synaptic input is likely still sufficient to reach threshold and drive firing. Thus, we were unable to perform this experiment. We have added a brief discussion in the methods to describe the limitations of this silencing method.

7. For recording experiments of the DCN, authors do not indicate the location of their recording relative to the rostral-caudal axis of the DCN. These segments have different RF and response properties. Post hoc histology or images of the recording sites should be included. Furthermore, for the lesion studies, the lesion location seems to be caudal part of DCN and part of the cervical spinal cord/ TG system. Have the authors lesioned across the entire DCN to make sure their lesion effects are not site specific. Lastly, for antidromic activation confirmation, images of electrode path would be useful.

Although in some species the DCN has a ‘core’ and ‘shell’ region, along with differences in properties along the rostro-caudal axis, we have not observed obvious differences in mice within the areas we recorded. Antidromically identified IC-PNs could be found among neighboring units with small receptive fields and other properties of VPL-PNs. We have included images of the extent of the gracile and images of an example recording site in Extended Data Figure 1, as well as images of stimulation sites in Extended Data Figure 1, as suggested. We have included more details in the methods about where in the DCN we collected our recordings.

For lesion studies, we lesioned the caudal part of the DCN that includes much of the gracile. We have found that lesioning this area also eliminates tactile-evoked DCN responses rostral to the site of the lesion, as many of the axons ascending the dorsal column will pass through this area to more rostral sites. Thus, we did not explore lesioning other parts of the gracile.

8. In the main text, the authors claim (line 153-154) that no CGRP expressing neurons contribute to the direct pathway to the DCN referring to a paper the lab previously published. A more precise statement should be made or additional references that support the claim should be included since the noted reference suggests there are sparse CGRP+ projections from DRG to DCN.

We have edited the text to note that there are very sparse CGRP inputs to the DCN, and the source is unknown. We have also included images of the DCN and neighboring trigeminal nucleus in *Calca-Flp; Rosa26^{FSF-ReaChR}* animals that we have used. This shows the high density of ReaChR-labeled axons in the trigeminal nucleus and, for comparison, very sparse labeling in the DCN. These data are shown in Extended Data Figure 8 of the revised manuscript. We have also included additional references to address sparse CGRP+ fibers in the DCN that have been observed. We address this topic further using physiological measures, as described in point 13 below, and found that while stimulating the skin of *Calca-Flp; Rosa26^{FSF-ReaChR}* mice evoked strong physiological responses in the DCN, providing light directly over the DCN evoked little or no responses.

9. To support the statement of line 166-167, graphs that show the percentage of units responding and not responding to mechanical stimuli should be shown.

We have included this information in Figure 4 of the revised manuscript.

10. The title is ambiguous and could just as well suggest that the authors will report that different areas of the DCN encode various features of touch. Similarly, the last sentence of the summary expresses concepts in a way that is so vague it makes them not surprising and further the descriptions of functional/biological purpose are also so vague that the essentially lack meaning and rather give the impression that not much was learned in the end from performing these analyses.

We respectfully disagree with the reviewer about whether the title is ambiguous. The title refers to dorsal column subdivisions, not the DCN, and adding more detail to the title would expand it beyond the character limit. We agree about the last sentence of the summary, and therefore we have changed it to better emphasize the main findings.

11. Line 46-49, would be good to make clear that the three DCN neuron types studied here comprise a majority of the neurons in the DCN but that other DCN PN populations exist (i.e spinal cord projecting neurons, PAG projecting neurons etc).

We have now added these projection neuron populations to the list of DCN cell types in the main text.

12. It would be helpful if there is a graphic representation that shows the quantified latencies of responses (line 159-160).

We have included a plot of quantified latencies as the reviewer has suggested in Figure 4 of the revised manuscript.

13. The phrasing used in the text: that activation of PN-VPL by optical activation in the skin did not show activation when optically activated in the DCN begs the question of whether optical activation of *Calca*⁺ afferents in DCN causes firing of any DCN neurons. I understand the prediction that if the PN-VPLs activated by skin are directly connected to *Calca*⁺ afferents from DRG, then you would expect they would be activated in DCN. However, any of the responses of DCN neurons to optical activation of *Calca*⁺ afferents in DCN should be reported.

We agree that this is an important point. We have thus performed additional experiments to ask how many DCN neurons can be activated by direct activation of any potential *Calca*⁺ fibers within the DCN. We found that very few DCN neurons can be activated directly by *Calca*⁺ endings in the DCN (2/19 randomly recorded units), and that these responses are very weak compared activation of all sensory neuron inputs over the DCN. These experiments are shown in Extended Data Figure 8, and we have included a brief discussion in the Extended Data Figure legend to explicitly state the prevalence of direct *Calca*⁺ inputs detected in electrophysiology experiments.

14. Why is the x-axis of Extended Data Fig. 7T shown as 100 mN max force but the figure legend discusses 300 mN?

As we performed this experiment, the maximum intensity delivered was 300 mN. However, we found if units were not activated by stimuli below 100 mN, they were never activated, and so the axis was truncated. We have changed the figure to make this more clear.

15. Line 465, in Extended Data Fig. 2, the title may be changed to projection neuron subtypes, not just IC PN since there are experiments addressing other subtypes. Also need more clear statement of what the experiment is. Was it only recorded from IC PN and then generalize the types to VPL PN? Or recorded from random neurons in the DCN?

We apologize that this figure was confusing, and we have modified it in an attempt to be clearer. The figure shows the different response profiles we observed among identified IC projection neurons. They fit into three categories: most are high frequency vibration-tuned, and the other two smaller groups are either spatially-tuned or contact-tuned. We used these same categories to classify identified VPL-PNs and found that almost all of them fall into the category of spatially-tuned.

16. Line 639, in Extended Data Fig. 7, there is a run-on sentence

We thank the reviewer for a careful reading of the manuscript. This sentence has been changed.

17. Line 702-703 mention that the corticospinal tract will be perturbed. Would be good to mention the same thing in Extended Data Fig. 7 and mention in the main text

We have added this to the main text and Extended Data Figure 7.

Referee #3 (Remarks to the Author):

The coding of tactile information has been extensively studied in the peripheral mechanoreceptors and in the primary somatosensory cortex. Recently, it has become clear that processing and integration of tactile information occurs at the intermediate target brain regions as primary mechanosensory information ascends the somatosensory neuroaxis. Mechanosensory signals for discriminative touch perception reach the brain via the spinal cord dorsal column directly from peripheral mechanoreceptors that innervate the dorsal column nuclei (DCN) of the brainstem. A second indirect pathway formed by post-synaptic dorsal column neurons (PSDCs) within the spinal cord dorsal horn, which receive input from mechanoreceptors, also project to the DCN. The significance and relative contributions of these two major DCN inputs to the coding of touch is still unclear. The work now submitted, importantly sheds light on how signals encoding different tactile features are differentially conveyed by the two independent pathways to converge in the DCN. The authors propose that the LTMRs and the direct dorsal column pathway underlie vibration tuning, whereas PSDC neurons and the indirect dorsal column convey the intensity of sustained stimuli, which constitutes an important and interesting contribution to the field.

Comments:

- After an initial characterization of the different DCN neuron types, where the authors describe that the most abundant ventral posterolateral thalamus VPL projecting neurons (VPL-PNs) have small excitatory receptive fields with large regions of surround suppression, opposed to inferior colliculus projecting neurons (IC-PNs) that have larger and exclusive excitatory receptive fields, they addressed how the unique response properties of DCN neuron subtypes are generated. Experiments shown in Figures 2A-J lead to the conclusion that the indirect pathway contributes to responses elicited by low-frequency (10Hz) mechanical stimuli, while high-frequency vibration (50 to 300Hz) is exclusively encoded by the direct dorsal column pathway. Since VPL-PNs seem to have a high degree of spatial information encoding due to their inhibitory surround receptive fields, the authors asked how the direct and indirect pathways contribute to this. Using a 50Hz vibration stimulation protocol, while selectively inhibiting the direct pathway, resulted in the suppression of the inhibitory surround of DCN neurons leading to the conclusion that the direct dorsal column pathway is the primary driver of inhibitory surround receptive fields in VPL-PNs. In order to test if the indirect pathway would also have a role in driving the inhibitory surround of VPL-PNs receptive fields, this reviewer suggests to perform the experiment 2K-L also with low frequency stimulation, since PSDC neurons do not seem to elicit a DCN response at high frequency stimulations. Also, as shown in Figure 3, PSDC neurons seem to mediate force intensity tuning in the DCN. I wonder if indentation stimulation is also able to elicit surround inhibition in DCN neurons. If so, I think it would be relevant to test this paradigm and explore the potential role of the indirect pathway to the spatial tuning of VPL-PNs.

We have performed new experiments to determine the contribution of PSDCs to inhibitory surrounds, using both low-frequency (10 Hz) stimulation and indentation to drive inhibition. We found that for both stimuli silencing the direct pathway almost completely eliminates these inhibitory surrounds in

most units. Thus, we think that PSDCs do not play a major role in spatial tuning or shaping the inhibitory surrounds of VPL-PNs. These findings are discussed on pages 5 and 10, and shown in Figure 2 of the revised manuscript.

We thank the reviewer for this suggestion, and we think the suggested experiments strengthened the paper.

- Figure 2H shows that silencing LTMR axon terminals to block direct pathway inputs, was not able to eliminate DCN responses upon 10Hz mechanical stimuli, leading to the conclusion that the indirect pathway was able to convey signals from low-frequency mechanical stimuli to generate responses in the DCN. This evidence is based on a relative low sample number with a p value of 0.056. Since this is a relevant information, I think this should be better corroborated with a more robust data set.

Thank you for making this point. We have performed additional experiments to make this a more robust data set. We found that responses to low-frequency stimuli were significantly decreased when silencing the direct pathway. However, there was a substantial response remaining when direct input was silenced, suggesting that both the direct and indirect pathways contribute to low-frequency stimuli.

- Figures 3D-M explore the contribution of indirect pathway to DCN representation of stimulus intensity. The authors nicely show that when excitatory transmission was blocked in the spinal cord, suppressing PSDC input, firing during the sustained phase of indentation was eliminated specially at high force ranges, pointing to an interesting observation that in the absence of PSDC input, DCN neurons are no longer able to encode the intensity of the maintained stimuli. To corroborate this observation I think this experiment should also be done when silencing the direct pathway (Acr1 experiment), as in this experiment only low forces stimuli were applied (5-10mN) and analysis such as shown in panels F-M should be included.

This experiment was also requested by reviewer 2, and we have copied our response from above here:

We initially set out to perform these experiments, but we were unable to adequately silence ascending input to the DCN when stimulating the skin at high forces. We delivered 100 mN indentation to skin in *Cdx2-Cre; Rosa26^{LSL-Acr}* animals in which all input to the DCN should be suppressed with light. However, 100 mN indentation activates such a powerful physiological response that it was capable of evoking some postsynaptic responses in DCN neurons even in the presence of light. In pilot experiments, we found that Acr was able to silence about 90% of synaptic release. When low-threshold stimuli were delivered, we think the remaining amount is incapable of evoking responses in the DCN. However, when we deliver intense stimuli, the remaining 10% of a large synaptic input is likely still sufficient to reach threshold and drive firing. Thus, we were unable to perform this experiment. We have added a brief discussion in the methods to describe the limitations of this silencing method.

Reviewer Reports on the First Revision:

Referees' comments:

Referee #1 (Remarks to the Author):

The authors have done a good job in addressing reviewer comments, providing clarifications, as well as including some new and important data. I have no further questions and recommend publication. This is a very nice study.

Referee #2 (Remarks to the Author):

The authors have satisfied my concerns.

Referee #3 (Remarks to the Author):

In response to my questions, the authors have provided additional experimental data which support their conclusions and strengthened their findings. In one case, the requested experiment was not possible and the authors explained the technical limitations. I am therefore in favor of publishing this manuscript without further changes.